# Neutralization Epitopes in Trimer and Pentamer Complexes Recognized by Potent Cytomegalovirus-Neutralizing Human Monoclonal Antibodies

Yuanbao Ai,[a,c] Changwen Wu,[a,b] Ming Zhang,[a,c] Dabbu Kumar Jaijyan,[e] Tong Liu,[a,b] Lipeng Zan,[a,c] Nan Li,[a,d] Wei Yu,[a,c] Yueming Wang,[a,b] Xiaohui Yuan,[a,b] Chengming Li,[a,b] Weihong Zheng,[b] Hua Zhu,[e] Hua-Xin Liao[a,b,c]

aDepartment of Cell Biology, College of Life Science and Technology, Jinan University, Guangzhou, China
bTrinomab Biotech Co., Ltd., Zhuhai, China
cInstitute of Biomedicine, Jinan University, Guangzhou, China
dGuangdong Provincial Key Laboratory of Bioengineering Medicine, Guangzhou, China
eDepartment of Microbiology and Molecular Genetics, New Jersey Medical School, Rutgers University, Newark, New Jersey, USA

Yuanbao Ai and Changwen Wu contributed equally to this work. Author order was determined on the basis of type of contribution.

**ABSTRACT** Human cytomegalovirus (HCMV) infects 36% to almost 100% of adults and causes severe complications only in immunocompromised individuals. HCMV viral surface trimeric (gH/gL/gO) and pentameric (gH/gL/UL128/UL130/UL131A) complexes play important roles in HCMV infection and tropism. Here, we isolated and identified a total of four neutralizing monoclonal antibodies (MAbs) derived from HCMV-seropositive blood donors. Based on their reactivity to HCMV trimer and pentamer, these MAbs can be divided into two groups. MAbs PC0012, PC0014, and PC0035 in group 1 bind both trimer and pentamer and neutralize CMV by interfering with the postattachment steps of CMV entering into cells. These three antibodies recognize antigenic epitopes clustered in a similar area, which are overlapped by the epitope recognized by the known neutralizing antibody MSL-109. MAb PC0034 in group 2 binds only to pentamer and neutralizes CMV by blocking the binding of pentamer to cells. Epitope mapping using pentamer mutants showed that amino acid T94 of the subunit UL128 and K27 of UL131A on the pentamer are key epitope-associated residues recognized by PC0034. This study provides new evidence and insight information on the importance of the development of the CMV pentamer as a CMV vaccine. In addition, these newly identified potent CMV MAbs can be attractive candidates for development as antibody therapeutics for the prevention and treatment of HCMV infection.

**IMPORTANCE** The majority of the global population is infected with HCMV, but severe complications occur only in immunocompromised individuals. In addition, CMV infection is a major cause of birth defects in newborns. Currently, there are still no approved prophylactic vaccines or therapeutic monoclonal antibodies (MAbs) for clinical use against HCMV infection. This study identified and characterized a panel of four neutralizing MAbs targeting the HCMV pentamer complex with specific aims to identify a key protein(s) and antigenic epitopes in the HCMV pentamer complex. The study also explored the mechanism by which these newly identified antibodies neutralize HCMV in order to design better HCMV vaccines focusing on the pentamer and to provide attractive candidates for the development of effective cocktail therapeutics for the prevention and treatment of HCMV infection.

**KEYWORDS** human cytomegalovirus, pentamer complex, gH/gL/gO, neutralizing antibodies, therapeutics, gH/gL reactive, pentamer specific, pUL128/UL130/UL131A

Address correspondence to Hua-Xin Liao, tliao805@jnu.edu.cn.

The authors declare no conflict of interest.

Human cytomegalovirus (HCMV) is a ubiquitous pathogen belonging to the *Betaherpesvirinae* subfamily of *Herpesviridae*, also known as human herpesvirus 5 (HHV5), which usually presents as asymptomatic infections mainly in immunocompetent populations and causes severe complications only in immunocompromised individuals. The prevalence of CMV infection in adults varies widely, ranging from 36% to 77% of the population in developed countries, while it is generally higher than 90% of the population in developing countries (1). Reinfection or reactivation of CMV in immunocompromised individuals can cause serious complications and even life-threatening situations (2, 3). In addition, congenital HCMV infection is a major cause of visual/hearing defects or mental retardation in 0.7% of the newborns (4, 5).

Membrane fusion of human herpesviruses is a complex process requiring the highly conserved envelope glycoproteins gB and gH/gL and multiple glycoprotein complexes to mediate attachment, fusion, or endocytosis of the virions to host cells (6, 7). The function of HCMV glycoprotein gB is currently considered to be a major fusogen mediating membrane fusion of virions with infected cells, which is triggered after trimer or pentamer complex binding to a receptor(s) (8–10). The two critical gH/gL-containing complexes encoded by HCMV are required for entry into cells. HCMV trimer complex is formed by gH/gL/gO, in which gL-Cys144 is linked with gO-Cys351 by disulfide bonds, and HCMV pentamer complex contains five different subunits, gH/gL/UL128/UL130/UL131A, in which gL-Cys144 binds to UL128-Cys162 by disulfide bonds. Because gO-Cys351 in trimer and UL128-Cys162 in pentamer share the same binding site with gL-Cys144, the formation of pentamer and the formation of trimer are mutually exclusive (11). In addition, the ratio of pentamer to trimer on the viral membrane surface is regulated by UL148 and US16 (12, 13), and it was recently shown that UL116 acts as a gH chaperone during the assembly and maturation of the gH complex in infected cells, also affecting the formation of the gH complex (14, 15). Genetic studies explicitly elucidated that pentamer is required for entry into epithelial and endothelial cells, but not into fibroblasts, and more importantly, UL128/UL130/UL131A in pentamer maintain HCMV tropism toward nonfibroblasts (16, 17). In contrast, trimer is considered to be sufficient for mediating CMV entry into fibroblasts. Several studies have shown that gO is required to maintain infectivity of cell-free virus and that gO-null virus does not infect either fibroblasts or epithelial/endothelial cells (18–20).

HCMV was first isolated in 1956 (21), and there are still no approved prophylactic vaccines or therapeutic monoclonal antibodies (MAbs) for clinical use against HCMV infection, although experimental vaccines and therapeutic MAbs have been in clinical trials. So far, CMV gB protein as the immunogen and MF59 as the adjuvant (gB/MF59) comprise the best-performing CMV vaccine candidate in clinical trials, with 43% to 50% protection in phase II clinical trials in solid-organ transplant (SOT) recipients (22). Chemical drugs such as ganciclovir (GCV), letermovir, maribavir (MBV), and CMV hyperimmune globulin (CMVIG) exhibit efficacy against HCMV (23–25). CMVIG shows promising results after SOT or allogeneic hematopoietic cell transplantation (26–28). Biweekly administration of CMVIG efficiently prevented maternal-fetal HCMV transmission after primary infection in the first trimester (29). However, current treatments are severely limited by toxicity or resistance to chemical drugs and by the drawbacks of blood-derived CMVIG products (30, 31). MAb-based therapy has advantages to overcome the limitations of the currently existing drugs and can be an effective clinical alternative. Pentamer-specific MAbs block HCMV infection of trophoblast progenitor cells (32), and increased levels of antibodies targeting pUL128L within 30 days after HCMV infection in pregnant women are associated with reduced risk of virus transmission to the fetus (33). Furthermore, MAbs with the ability to neutralize CMV in epithelial cells can protect SOT recipients (34). Meanwhile, gH-specific MAbs have shown a broad-spectrum characteristic to inhibit virus infection and transmission (35). A recent phase II clinical trial demonstrated that vaccination with Genentech's RG7667 (which includes pentamer-specific and anti-gH antibodies) delayed the time to CMV viremia and was associated with less CMV disease than the placebo in high-risk kidney transplant recipients (36). Thus, antibody cocktail therapy comprising pentamer-specific and gH-specific

MAbs may lead to a breakthrough in immunotherapy for CMV. However, currently, there are still no approved prophylactic vaccines or therapeutic MAbs for clinical use against HCMV infection.

In this study, we identified and characterized a panel of eight HCMV pentamer-reactive MAbs that can be divided into two groups based on their reactivity to CMV trimer and pentamer. MAbs PC0004, PC0010, PC0012, PC0014, PC0035, and PC0037 in group 1 bind both trimer and pentamer. We demonstrated that three (PC0012, PC0014, and PC0035) of six MAbs in group 1 neutralize HCMV and recognize a highly conserved domain on gH/gL proteins in trimer and pentamer. These MAbs neutralized HCMV not by blocking the binding of CMV to host cells but rather by inhibiting the postadsorption process of virus entry into the host cells. MAbs PC0031 and PC0034 in group 2 bind only to pentamer. We found that one (PC0034) of two pentamer-specific MAbs neutralizes HCMV by blocking virus adsorption by the host cells. Further analysis demonstrated that MAb PC0034 targets an antigenic site that is 100% conserved in UL128 and UL131A proteins in a total of 214 CMV genome sequences currently collected in the NCBI database.

Taken together, the findings of our study revealed key residues on the antigenic epitopes targeted by these two groups of potent neutralizing antibodies and provided valuable information on neutralizing epitopes on trimer or pentamer complexes for designing and developing trimer and/or pentamer as a CMV vaccine. The potent neutralizing MAbs identified in this study may be attractive candidates for the development of a therapeutic antibody "cocktail" for the prevention and treatment of HCMV infection.

## RESULTS

**Isolation of CMV pentamer-reactive antibodies from two individuals.** CMV pentamer and trimer complexes were produced recombinantly in HEK293F cells by transient transfection, isolated initially by Ni-nitrilotriacetic acid (Ni-NTA) agarose affinity chromatography, further purified by size exclusion chromatography (SEC) (Fig. 1A), and confirmed by SDS-PAGE analysis (Fig. 1B). Blood specimens from 50 volunteers were screened by enzyme-linked immunosorbent assay (ELISA) to identify samples with high binding titers with the purified recombinant pentamer (Fig. 2A). Human peripheral blood mononuclear cells (PBMCs) of four specimens with the highest antibody binding titers were selected for sorting single pentamer-reactive memory B cells by flow cytometry using dual-color fluorescence-labeled pentamer as a probe.

Immunoglobulin (Ig) variable regions of heavy- and light-chain gene segments ($V_H D J_H$ and $V_L J_L$) were amplified by reverse transcription-nested PCR from the sorted single B cells (Fig. 2B) and used to produce recombinant MAbs by use of a method described previously (37). A total of 16 recombinant antibodies were identified from the sorted single B cells from only two of four samples to bind the recombinant CMV pentamer (Fig. 2C). The somatic mutations of the identified pentamer-reactive antibodies range from 2.01% to 15.38% for $V_H$ genes and 6.47% to 19.93% for $V_L$ genes. These 16 antibodies were derived from 11 different $V_H$ gene clonal lineages, 8 of which (PC0004 and PC0031 from sample Es0050 and PC0010, PC0012, PC0014, PC0034, PC0035, and PC0037 from sample Es0079), representing 8 different clonal lineages, were selected for production of purified antibodies for further characterization (Table 1).

**Reactivity with recombinant CMV trimer and pentamer complexes.** To identify the protein components in CMV pentamer complex recognized by the isolated pentamer-reactive antibodies, recombinant CMV trimer and pentamer complexes were produced, purified (Fig. 1A), and verified by the known CMV gH/gL-reactive and pentamer-specific neutralizing antibodies (Fig. 3). Eight purified antibodies were tested in ELISA to bind recombinant CMV trimer and pentamer complexes using MAbs 8I21 and 9I6, known as CMV pentamer-specific MAbs, and MSL-109, known as a gH/gL-reactive MAb, as controls. We found that the tested eight MAbs can be divided into two groups based on their reactivity to CMV trimer and pentamer. Antibodies PC0004, PC0010, PC0012, PC0014, PC0035, and PC0037 in group 1 bound both pentamer and trimer with a binding profile similar to that of the known HCMV-neutralizing MAb MSL-109 (38), which was reported to bind gH

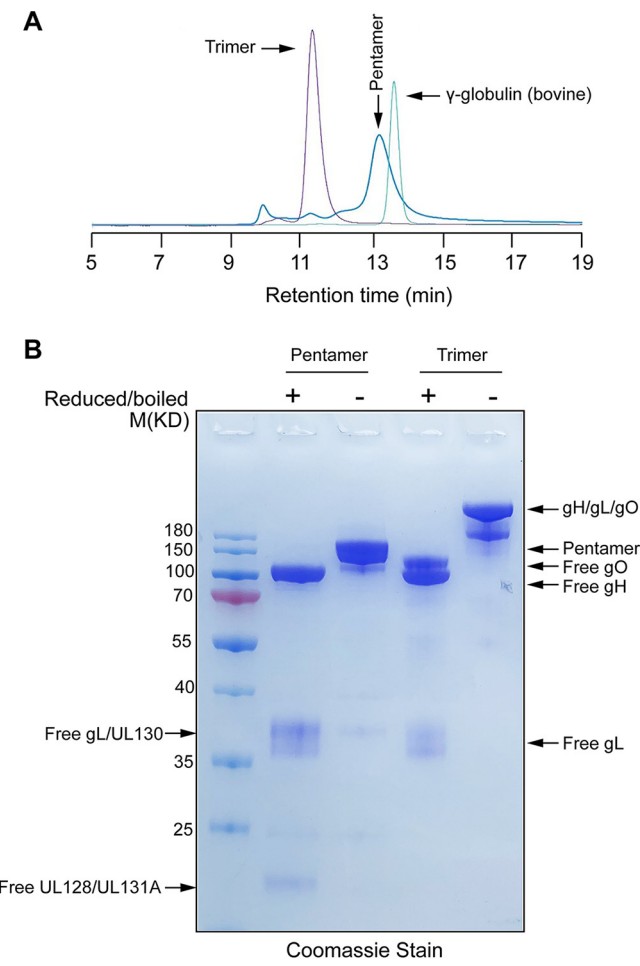

**FIG 1** Analysis of purified recombinant gH/gL/UL128/UL130/UL131A pentamer and gH/gL/gO trimer complexes. (A) Size exclusion chromatography profiles of the indicated recombinant gH/gL/UL128/UL130/UL131A pentamer, gH/gL/gO trimer, and gamma globulin with a molecular weight (MW) of 158 kDa as a size reference. Both pentamer and trimer complexes are shown as nearly symmetrical peaks, indicating their high purity. (B) The major peaks of pentamer and trimer complexes in the plot in panel A were collected and fractionated in parallel with MW standards in a 4% to 20% gel by SDS-PAGE under reduced (+) and nonreduced (−) conditions. Gels were stained with Coomassie blue. CMV pentamer and trimer subunit protein bands and molecular weight standards are indicated.

subunit in CMV trimer and pentamer complexes, although PC0004 bound CMV trimer with weaker apparent binding affinity (Fig. 3). In contrast, PC0031 and PC0034 in group 2 bound only to CMV pentamer and did not to bind to trimer, just like the known HCMV pentamer-specific neutralizing MAbs 8I21 and 9I6 (39), which bind only to CMV pentamer and not to trimer (Fig. 3). Since there are gH/gL in both CMV trimer and pentamer complexes, these results suggested that MAbs in group 1 likely recognize the gH/gL in the complexes, while two MAbs in group 2 may recognize the pUL128/130/131A (Fig. 3).

**Neutralization activity of pentamer-reactive human MAbs.** All eight purified MAbs were tested for the ability to neutralize CMV in human fetal lung fibroblast cells (MRC-5). We found that of six antibodies in group 1, three MAbs, PC0012, PC0014, and PC0035, along with the control MAb MSL-109 neutralized HCMV laboratory standard Towne strain with 50% effective concentrations ($EC_{50}$s) of 0.648 to 0.938 $\mu$g/mL and also neutralized the HCMV wild-type strain BE13/2012 with $EC_{50}$ of 0.299 to 1.480 $\mu$g/mL (Fig. 4A and B). Two pentamer-specific MAbs, PC0031 and PC0034, in group 2 did not neutralize either CMV Towne strain (Fig. 4A) or clinical isolate BE13/2012 (data not shown) in fibroblast cell-based neutralization assays, just like the known pentamer-specific neutralizing antibodies 8I21 and 9I6 (39). However, of two pentamer-specific antibodies, only

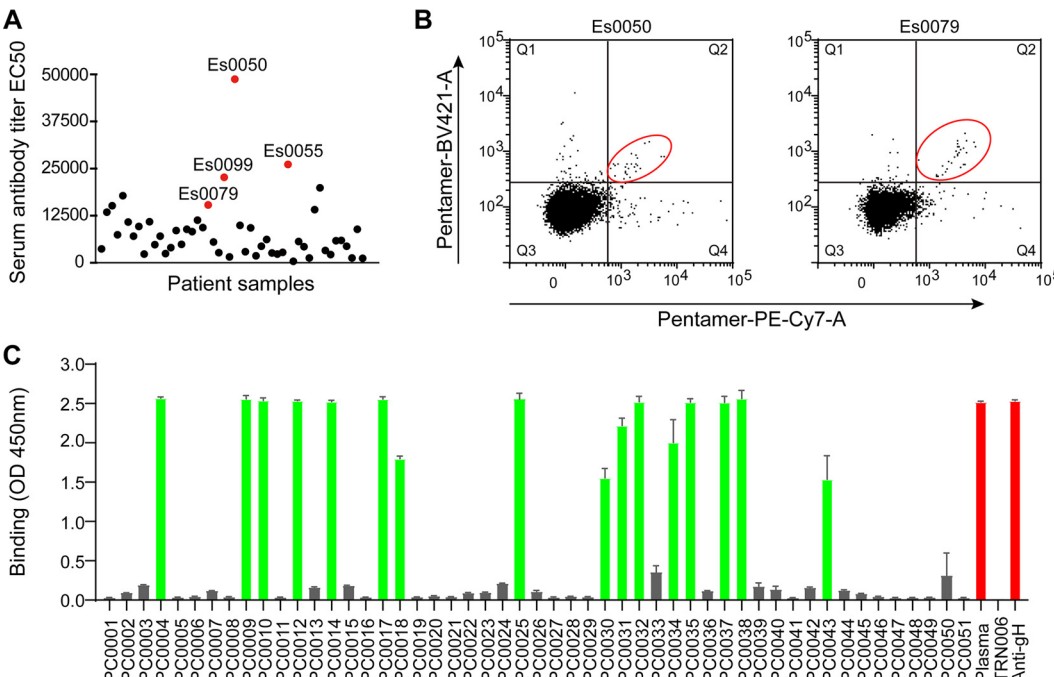

**FIG 2** Isolation of HCMV pentamer-reactive human MAbs. (A) Serum samples were screened to identify the samples with high binding titers ($EC_{50}$) to CMV pentamer in ELISAs. Four serum samples with high serum pentamer binding titers are highlighted with solid red circles. (B) Flow cytometry analysis of CMV pentamer-specific memory B cells (circled in red) identified by dual color-labeled CMV pentamer as a probe by using a FACS Aria III from two (Es0050 and Es0079) of four PBMC samples. Data for the other two samples (Es0055 and Es0099) used for sorting are not shown, because no CMV pentamer-reactive antibodies were isolated from these donors. (C) CMV pentamer-reactive recombinant antibodies derived from the sorted single B cells were identified by ELISA. Cultures of 293T cells were transfected with the Ig heavy- and light-chain linear expression cassettes assembled from the isolated $V_H D_H J_H$ and $V_L J_L$ derived from sorted single B cells. Supernatants from the transfected cultures in triplicate were screened for CMV pentamer-reactive recombinant antibodies. CMV-positive plasma and a commercial anti-gH MAb were used as positive controls. Antigenic specificity-irrelevant antibody TRN006 was used as a negative control. Columns in green color represent MAbs with strong binding reactivity to pentamer.

PC0034 was tested in ARPE-19 epithelial cell-based neutralization assays and was demonstrated to neutralize two tested HCMV wild-type strains, VR1814 and NR, and one rescued HCMV strain, AD169 FIX (laboratory strain with reconstituted pentamer expression), with $EC_{50}$s of 0.068, 0.070, and 0.079 $\mu$g/mL, respectively (Fig. 4C).

**TABLE 1** V(D)J rearrangement of the isolated CMV trimer/pentamer-reactive antibodies

| MAb[a] | $V_H$ | | | | | | $V_L$ | | | |
|---|---|---|---|---|---|---|---|---|---|---|
| | V | D | J | % Mutation | CDR3 | Isotype | $\kappa/\lambda$ | J | % Mutation | CDR3 |
| PC0009 | 4-39 | 3-10 | 5 | 15.38 | 15 | G3$\kappa$ | 2-28 | 3 | 8.02 | 9 |
| **PC0010** | 4-39 | 3-10 | 5 | 14.58 | 15 | G3$\kappa$ | 2-28 | 3 | 7.36 | 9 |
| PC0017 | 4-39 | 3-10 | 5 | 15.18 | 15 | G3$\kappa$ | 2-28 | 3 | 7.41 | 9 |
| PC0025 | 4-39 | 3-10 | 5 | 12.80 | 15 | G3$\kappa$ | 2-28 | 3 | 6.79 | 9 |
| **PC0034** | 3-23 | 3-9 | 4 | 11.48 | 20 | G1$\lambda$ | 3-21 | 2 | 9.39 | 11 |
| PC0043 | 3-23 | 3-9 | 4 | 11.45 | 20 | G1$\lambda$ | 3-21 | 2 | 9.35 | 11 |
| **PC0037** | 5-51 | 1-20 | 4 | 11.60 | 13 | G3$\lambda$ | 3-25 | 2 | 11.69 | 11 |
| PC0038 | 5-51 | 1-20 | 4 | 11.60 | 13 | G3$\lambda$ | 3-25 | 2 | 12.26 | 11 |
| PC0018 | 3-30 | 1-7 | 5 | 9.58 | 12 | G1$\kappa$ | 1-39 | 2 | 19.93 | 9 |
| PC0032 | 3-30 | 5-12 | 6 | 2.01 | 20 | M | 2-11 | 3 | 8.97 | 10 |
| **PC0012** | 7-4 | 3-16 | 4 | 14.29 | 12 | G1$\kappa$ | 1-5 | 1 | 9.74 | 9 |
| **PC0035** | 7-4 | 3-10 | 5 | 13.47 | 20 | G3$\lambda$ | 2-14 | 1 | 7.92 | 10 |
| **PC0014** | 4-34 | 2-2 | 3 | 14.59 | 14 | G3$\kappa$ | 1-39 | 2 | 6.47 | 9 |
| **PC0004** | 3-43 | 5-12 | 6 | 13.16 | 19 | G1$\kappa$ | 2-28 | 4 | 10.74 | 9 |
| PC0030 | 3-30 | 6-19 | 5 | 11.94 | 15 | G1$\lambda$ | 2-14 | 3 | 6.62 | 10 |
| **PC0031** | 3-33 | 5-24 | 6 | 12.97 | 20 | G3$\lambda$ | 3-25 | 3 | 17.68 | 11 |

[a]The eight representative pentamer-reactive antibodies are highlighted in bold.

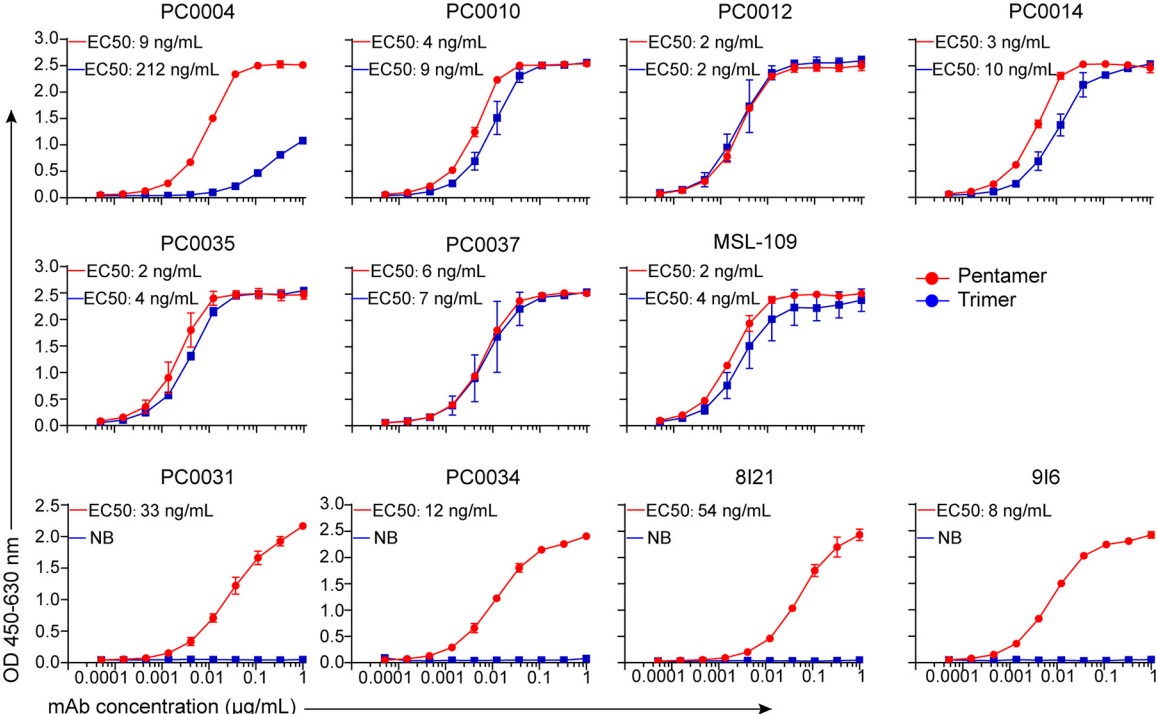

**FIG 3** Binding specificity of the isolated antibodies to pentamer and trimer proteins. The indicated antibodies (including the control antibodies) in serial 3-fold dilutions were tested for binding to pentamer (red curves) and trimer (blue curves). EC$_{50}$ values for binding of the individual MAbs to pentamer in red curves and trimer in blue curves are indicated. NB, no binding. Data are shown as the mean ± standard deviation (SD) of results of three independent assays.

**Binding and cross-reactivity of neutralizing antibodies.** Binding affinities of neutralizing antibodies in these two groups of HCMV-neutralizing antibodies to the purified gH/gL/gO trimer and the gH/gL/UL128/UL130/UL131A pentamer were measured and compared with those of the known neutralizing antibody MSL-109 by surface plasmon resonance (SPR). Three gH/gL-reactive neutralizing MAbs, PC0012, PC0014, and PC0035, bound to pentamer with high affinity at an affinity constant ($K_D$) of $2.85 \times 10^{-10}$ M, $3.23 \times 10^{-10}$ M, and $1.83 \times 10^{-10}$ M (Fig. 5A), respectively, and bound to trimer with affinity at a $K_D$ of $9.38 \times 10^{-11}$ M, $1.00 \times 10^{-9}$ M, and $1.01 \times 10^{-10}$ M (Fig. 5B), respectively. The binding affinities of MAbs PC0012, PC0014, and PC0035 to pentamer are approximately 4- to 7-fold higher than that of MAb MSL-109 (Fig. 5A and Table 2), and those to trimer are approximately 20- to 230-fold higher than that of MAb MSL-109 (Fig. 5B and Table 2).

To gain insight into the relationship of the antigenic epitope(s) recognized by MAbs PC0012, PC0014, and PC0035 in comparison with the known neutralizing antibody MSL-109, cross-competitive blocking binding of these antibodies to CMV pentamer was determined by SPR and ELISA (Fig. 5C and 6). Using ELISAs, we found that no MAbs in group 1 compete with MAbs in group 2 for binding and vice versa (Fig. 6). Among the three newly isolated gH/gL-reactive neutralizing antibodies in group 1, PC0012 blocked most, if not all, of the binding of MAbs PC0014 and PC0035 to CMV pentamer, while PC0014 only partially blocked, and PC0035 did not at all block, the binding of PC0012 to CMV pentamer. PC0014 and PC0035 did not block the binding of each other to CMV pentamer (Fig. 5C and 6). PC0012 and PC0035 completely blocked the binding of MSL-109 to CMV pentamer, while MSL-109 did not block the binding of PC0012 to CMV pentamer but could block the binding of PC0035 to CMV pentamer. PC0014 and MSL-109 did not mutually block the binding to CMV pentamer (Fig. 5C and 6).

It has been shown that among the MSL-109-resistant mutants, W168C/R, P171H/S, and D446N mutations in gH subunit contributed to the neutralization resistance to MSL-

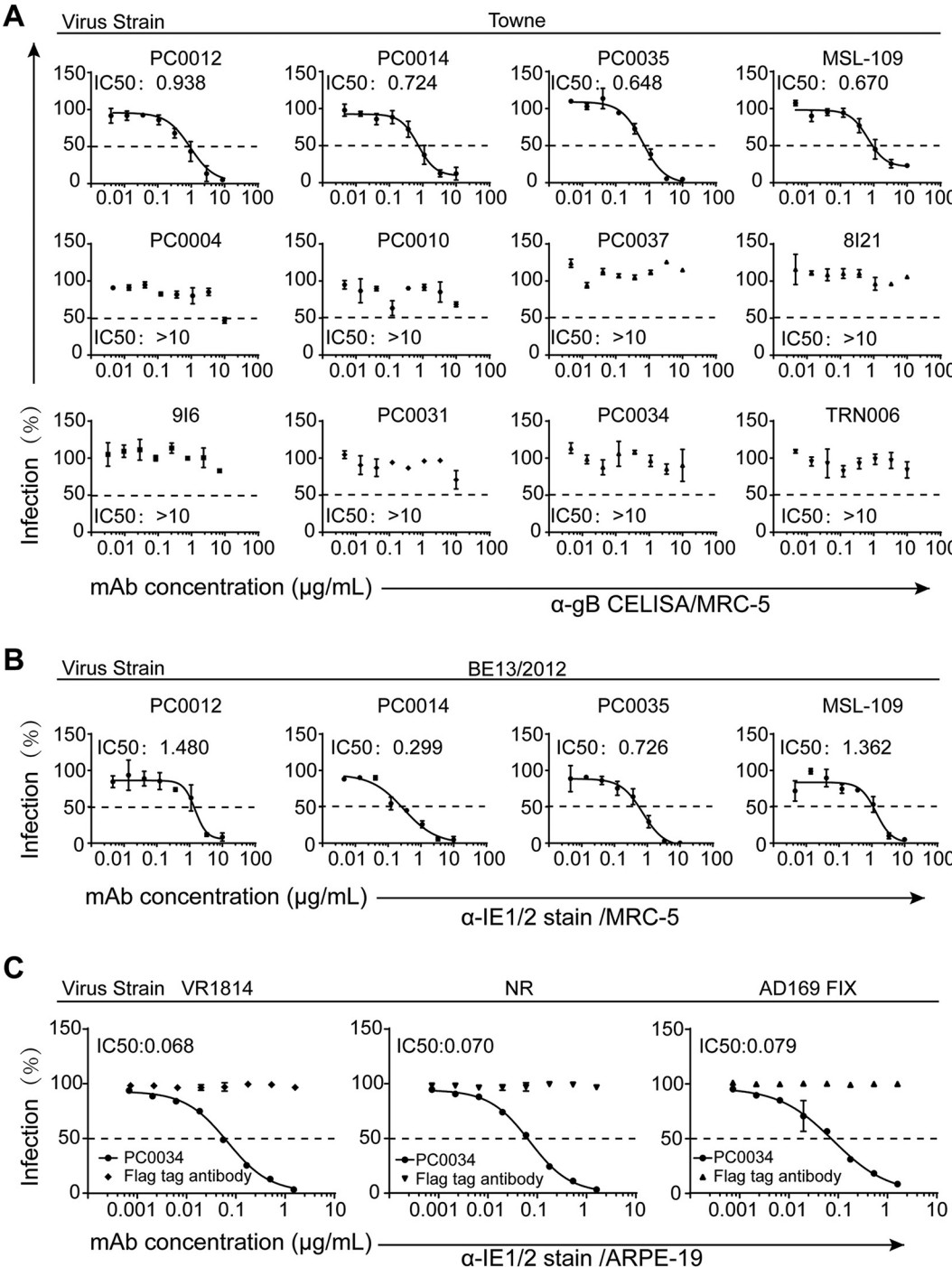

**FIG 4** Neutralization activity of the isolated pentamer-reactive MAbs. Pentamer-reactive MAbs, including positive-control MAbs and negative-control MAb TRN006, as indicated above the individual plots in eight 3-fold serial dilutions at the starting concentration of 10 μg/mL and PC0034 at the starting concentration of 1.6 μg/mL were tested for neutralization activity against CMV laboratory standard strain Towne in MRC-5 cells (A), wild-type strain BE12/2013 in MRC-5 cells (B), and HCMV strains VR1814, NR, and AD169 FIX in ARPE-19 cells (C). The percent inhibition (± standard error) of CMV infection (y axis) by the indicated antibodies as a function of antibody concentration (x axis) was plotted with data from the duplicated wells. The representative result of two independent experiments is shown. IC$_{50}$ values in micrograms per milliliter for each of individual antibodies are shown in the individual plots.

109, while MSL-109 can still bind gH/gL with the P171 (P171H/S) and D446 mutations (38). To further evaluate whether MAbs PC0012, PC0014, and PC0035 were affected by MSL-109-resistant mutations, we produced three pentamer mutants, each with a single amino acid substitution with alanine at the position corresponding to that of MSL-109-

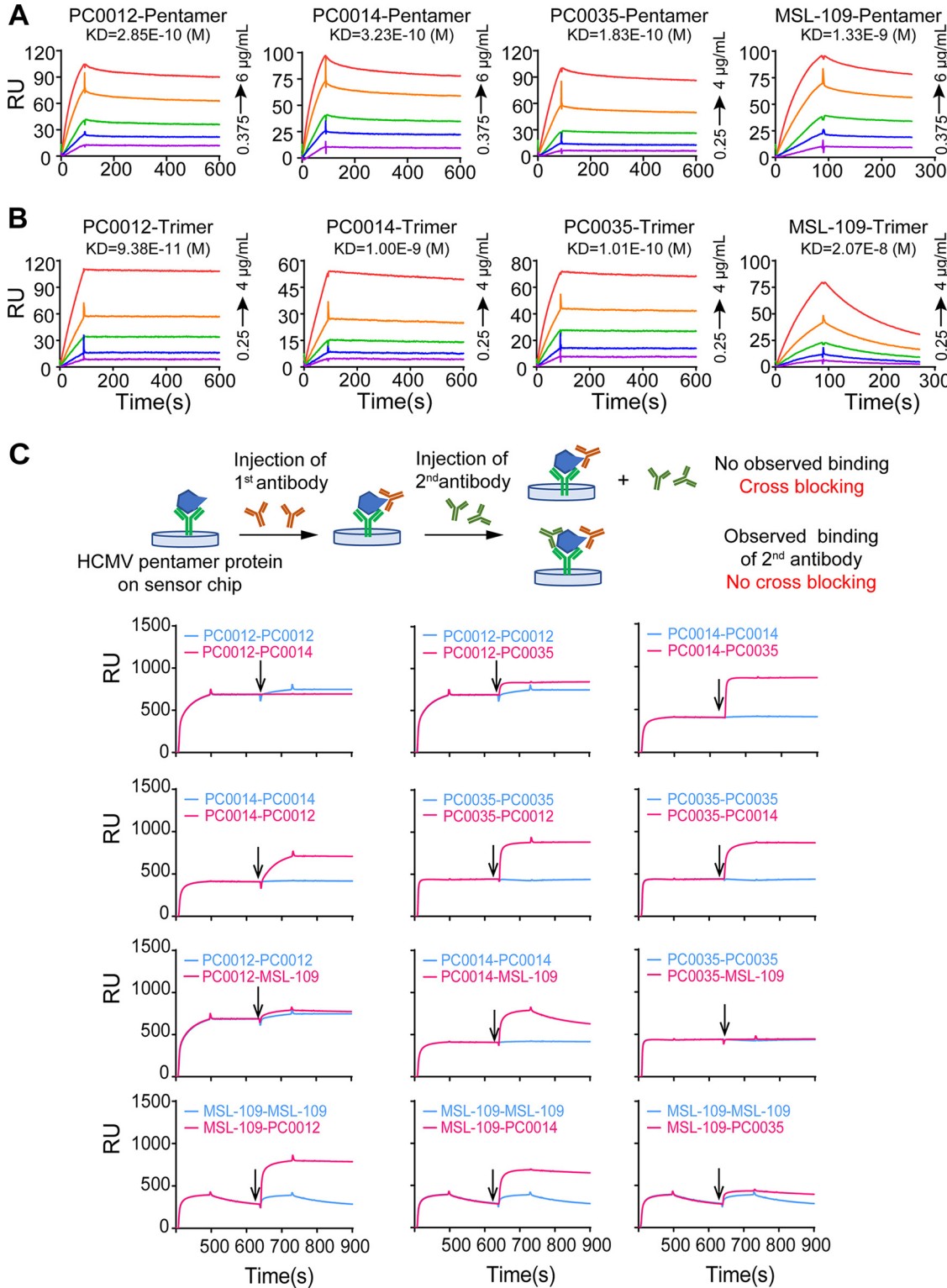

**FIG 5** Binding affinity and cross-reactivity of MAbs PC0012, PC0014, and PC0035. Affinity of MAbs PC0012, PC0014, and PC0035 as well MSL-109 to pentamer (A) and trimer (B) were measured by SPR. Each of the tested antibodies at five different concentrations, as indicated in the individual plots, was injected and allowed to flow over CMV pentamer (A) or trimer (B) captured on a Biacore chip for the length of time in seconds indicated at the bottom. Kinetic rate constants and $K_D$ were derived as described in Materials and Methods. Data are representative of at least two measurements. (C) Cross-blocking of the binding of antibodies to CMV pentamer was assayed by SPR, as illustrated in the schematic diagram. The first tested antibodies were injected and allowed to flow over CMV pentamer captured on a chip, followed by injection of the secondary antibody (indicated by an arrow). Overlap of the two binding curves in red and blue indicates that the binding to pentamer of the secondary antibody was blocked by the first antibody. In contrast, an increase in the binding curves after injection of secondary antibody indicates that the binding to pentamer of the secondary antibody was not blocked by the first antibody.

**TABLE 2** Binding affinity of MAbs PC0012, PC0014, and PC0035 to CMV pentamer and trimer[a]

| gH/gL complex | Antibody | $K_a \times 10^4$ (M$^{-1}$ s$^{-1}$) | $K_d \times 10^{-4}$ (s$^{-1}$) | $K_D$ (nM) | Fold difference in $K_D$ |
|---|---|---|---|---|---|
| Pentamer | PC0012 | 93.86 | 2.67 | 0.28 | 4.75 |
| | PC0014 | 101.80 | 3.29 | 0.32 | 4.16 |
| | PC0035 | 356.20 | 6.53 | 0.18 | 7.39 |
| | MSL-109 | 117.30 | 15.58 | 1.33 | |
| Trimer | PC0012 | 30.02 | 0.28 | 0.09 | **230.22** |
| | PC0014 | 19.10 | 1.92 | 1.00 | 20.72 |
| | PC0035 | 75.69 | 0.77 | 0.10 | **207.20** |
| | MSL-109 | 47.63 | 98.71 | 20.72 | |

[a]Boldface values indicate the fold change in affinity, the ratio of the affinity constant of the MSL-109 to trimer to the affinity constant of MAbs PC0012, PC0014 and PC0035 to trimer.

resistant mutants, including gH_W168A, gH_P171A, and gH_D446A. We found that all of these newly isolated neutralizing MAbs bound strongly to these mutants (Fig. 7A), while MAb MSL-109 did not bind or bound weakly to these mutants (Fig. 7A). Taken together, these results indicated that MAbs PC0012, PC0014, and PC0035 recognize the antigenic epitopes clustered in a similar area or region which are different from, but partially overlapping with or near to, the epitope recognized by the known neutralizing antibody MSL-109. In addition, antibody PC0012 recognizes a much broader region than do the other two antibodies and the well-defined CMV neutralizing antibody MSL-109.

**Mechanism of neutralization of CMV by gH/gL-reactive MAbs.** It has been reported that anti-gH/gL and anti-gB MAbs neutralize CMV at postattachment entry

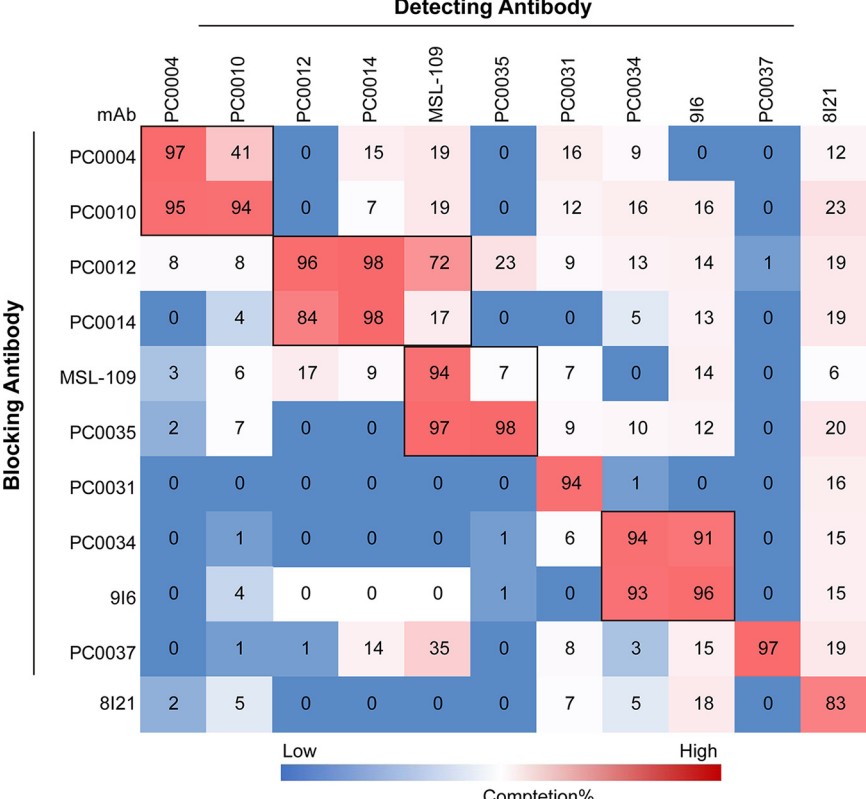

**FIG 6** Cross-blocking of the binding of pentamer-reactive MAbs to CMV pentamer. A heat map shows numbers in the individual squares representing the percentages of inhibition of the binding of detecting antibody (shown at the top) to pentamer protein by the blocking antibody (shown on the left) in ELISAs. Higher percentages of inhibition are represented by darker red color.

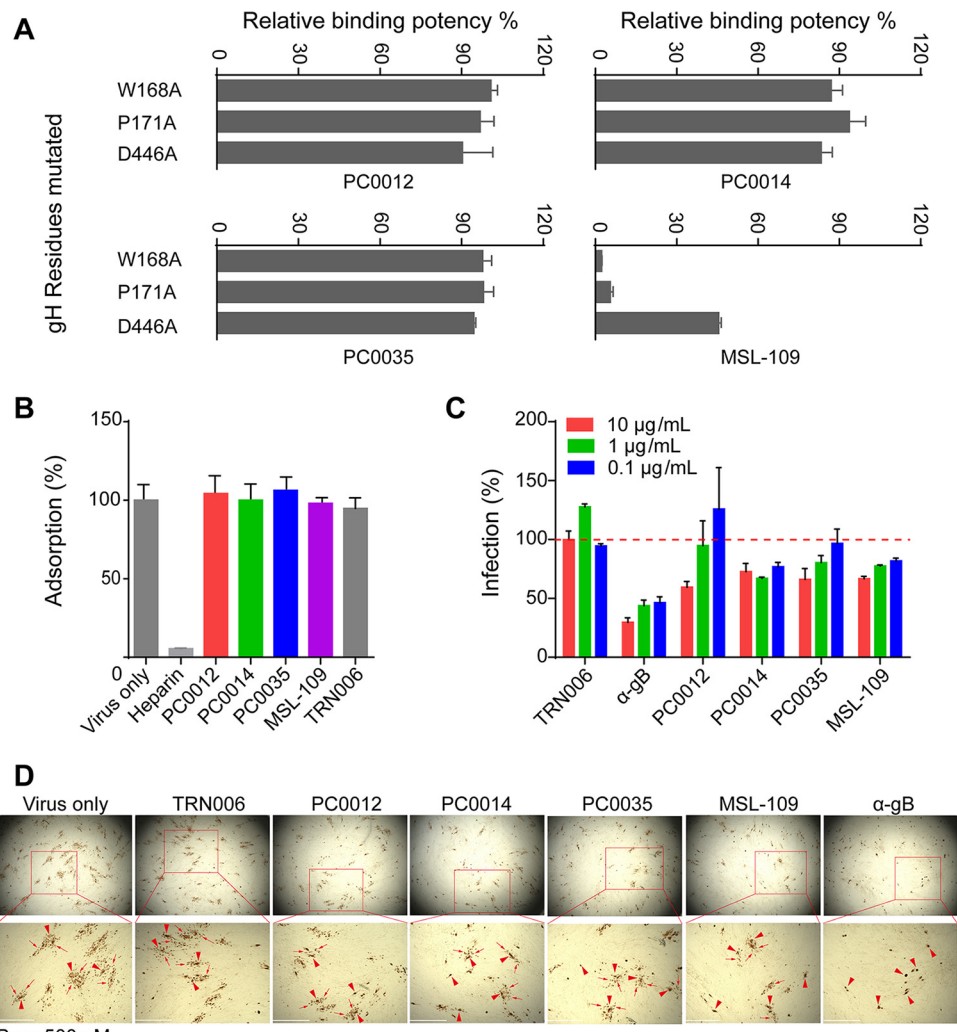

**FIG 7** Functional role of MAbs PC0012, PC0014, and PC0035 in the neutralization of CMV. (A) Binding of MAbs PC0012, PC0014, PC0035, and MSL-109 to pentamer mutants gH_W168A, gH_P171A, and D446A were evaluated in ELISAs. Relative binding potency is the ratio of the binding value of MAbs PC0012, PC0014, PC0035, and MSL-109 to pentamer mutants to that to wild-type pentamer. (B) The effect of preincubation of neutralizing MAbs PC0012, PC0014, and PC0035 with CMV on the adsorption of CMV onto the surface of MRC-5 cells was evaluated in CMV adsorption assays and quantified by quantitative PCR (qPCR) as described in Materials and Methods. Heparin, known to inhibit CMV adsorption to cells, was used as a positive control. TRN006 was used as a negative control. MSL-109 is a reference control. Shown are percentages (y axis) of change in virus copies affected by each of the tested samples (x axis) in comparison with the CMV viral copies measured in the virus-only group. (C) The effect of PC0012, PC0014, and PC0035 alone with a negative-control antibody (TRN006), anti-gB positive-control antibody (α-gB), and a reference antibody (MSL-109) at concentrations of 10 μg/mL, 1 μg/mL, and 0.1 μg/mL on CMV infectivity after the attachment of virus to cells was evaluated in postattachment neutralization assays as described in Materials and Methods. Shown are percentages of change in CMV infection affected by individual antibodies (x axis) in comparison with the infection measured in the negative-control antibody group. (D) The effect of PC0012, PC0014, and PC0035 alone with virus control, TRN006 negative control, anti-gB positive control (α-gB), and reference antibody (MSL-109) at concentrations of 50 μg/mL on cell-to-cell spread of HCMV in MRC-5 cells was evaluated in HCMV spread assays as described in Materials and Methods. IE1/IE2-positive HCMV-infected cells stained a brown color. The smaller brown dots indicated by arrows in the enlarged pictures (white bar, 500 μm) represent IE1/IE2-positive satellite viral replication centers resulting from transmission of HCMV from the initial viral replication centers, which are shown as slightly bigger, darker brown dots (indicated by arrowheads) in the initially infected cells.

steps (35, 40). To elucidate which step of the process in CMV infection is affected by MAbs PC0012, PC0014, and PC0035, we first tested the effect of preincubation of MAbs PC0012, PC0014, and PC0035 with CMV Towne strain on the attachment of CMV to the cultured MRC-5 cells at 4°C, under which condition CMV attachment is permitted and

the entry process is frozen. We found that preincubation of MAbs PC0012, PC0014, and PC0035 with CMV Towne strain using MSL-109 as a reference antibody and TRN006 as a negative-control antibody did not result in the reduction of CMV viral loads, as measured by real-time quantitative PCR, while preincubation of heparin as a positive control (41) resulted in detection of almost no viral DNA copies (Fig. 7B). Subsequently, we examined whether PC0012, PC0014, and PC0035 can affect CMV infection after attachment of CMV to cells. CMV Towne strain was preinoculated into MRC-5 cells for 30 min at 4°C before the MAbs were added. We found that PC0012, PC0014, and PC0035 along with MAb MSL-109 all reduced CMV infection under this condition (Fig. 7C).

HCMV infects cells also through cell-to-cell spread, resulting in syncytia (42). We evaluated the effect of MAbs PC0012, PC0014, and PC0035 on the cell-to-cell spread of HCMV in HCMV cell-to-cell spreading assays. HCMV infection was detected through CMV IE1/IE2 expression 5 days after virus inoculation. In comparison with the larger size and greater number of brown-stained satellite replication centers spreading from the initially infected cells seen in the virus-infected cells without the addition of antibody (virus only) or treated with negative control antibody TRN006, evenly dispersed brown-stained satellite viral replication centers of smaller size and fewer numbers could be observed in the infected cells treated with MAb PC0012, PC0014, PC0035, or MSL-109 (Fig. 7D). In contrast, only sporadic individual brown-stained replication centers without satellite viral replication centers could be seen in the infected cells treated with anti-gB neutralizing control antibody (Fig. 7D). These results indicate that antibodies PC0012, PC0014, and PC0035 along with MSL-109 at 50 $\mu$g/mL had an inhibitory effect on cell-to-cell spread of HCMV Towne in fibroblasts, while anti-gB neutralizing antibody had a substantially stronger inhibitory effect on cell-to-cell spread of HCMV, which is consistent with previously reported results (43).

Together, these results demonstrated that MAbs PC0012, PC0014, and PC0035 neutralize CMV through a mechanism by which binding is not blocked but which likely interferes in the postattachment steps of virus entry into cells.

**PC0034 blocks the binding of pentamer to epithelial/endothelial cells.** CMV-neutralizing MAb 9I6 (39) was well defined to recognize pentamer-specific site 5 and blocks CMV pentamer binding to cell surface receptor Nrp2 (44, 45). MAb PC0034 bound to CMV pentamer with affinity at a $K_D$ of $1.05 \times 10^{-10}$, while MAb 9I6 bound to pentamer with a slightly lower affinity, at a $K_D$ of $4.84 \times 10^{-10}$ M, than that of PC0034 (Fig. 8A). SPR analysis has demonstrated that MAb PC0034 completely blocks the binding of MAb 9I6 to CMV pentamer and vice versa (Fig. 8B). To use recombinant pentamer as a surrogate for assessing the ability of MAb PC0034 to block the binding of CMV to endothelial and epithelial cells, we first verified if recombinant pentamer can bind to endothelial and epithelial cells in cell ELISAs (CELISAs). We found that purified recombinant pentamer indeed bound to both epithelial cells (APRE-19) and human umbilical vein endothelial cells (HUVECs) in a dose-response relationship (Fig. 8C) that is consistent with previously reported observations (46–48). Preincubation of MAb PC0034 with CMV pentamer inhibited the binding of the pentamer to both epithelial and endothelial cells (Fig. 8D). In contrast, MAb MSL-109 or the negative-control MAb TRN006 did not affect the binding of the pentamer to these two cell types (Fig. 8D). Thus, these data indicate that MAb PC0034 neutralizes CMV by binding to the pentamer epitope that is critical for CMV attachment to the host cell receptor(s).

**Binding epitope recognized by PC0034.** To determine the epitope of PC0034, based on the epitopes recognized by antibody 9I6 (47) and indicated by molecular simulation analysis, a total of 11 pentameric variants, including 4 variants with a single amino acid mutation in the UL131 subunit and 7 variants with a single amino acid mutation in the UL128 subunit, were produced and purified. The integrity of the mutant pentamer variants was first assessed by ELISA using MAbs PC0004 and PC0035 as internal controls, since these two antibodies bind to both trimer and pentamer but are not neutralizing. We found that these 11 variants as well as the wild-type pentamer could be bound equally well by both antibodies PC0004 and PC0035, suggesting that there are no apparent quality differences among these 11 pentamer variants (Fig. 9A

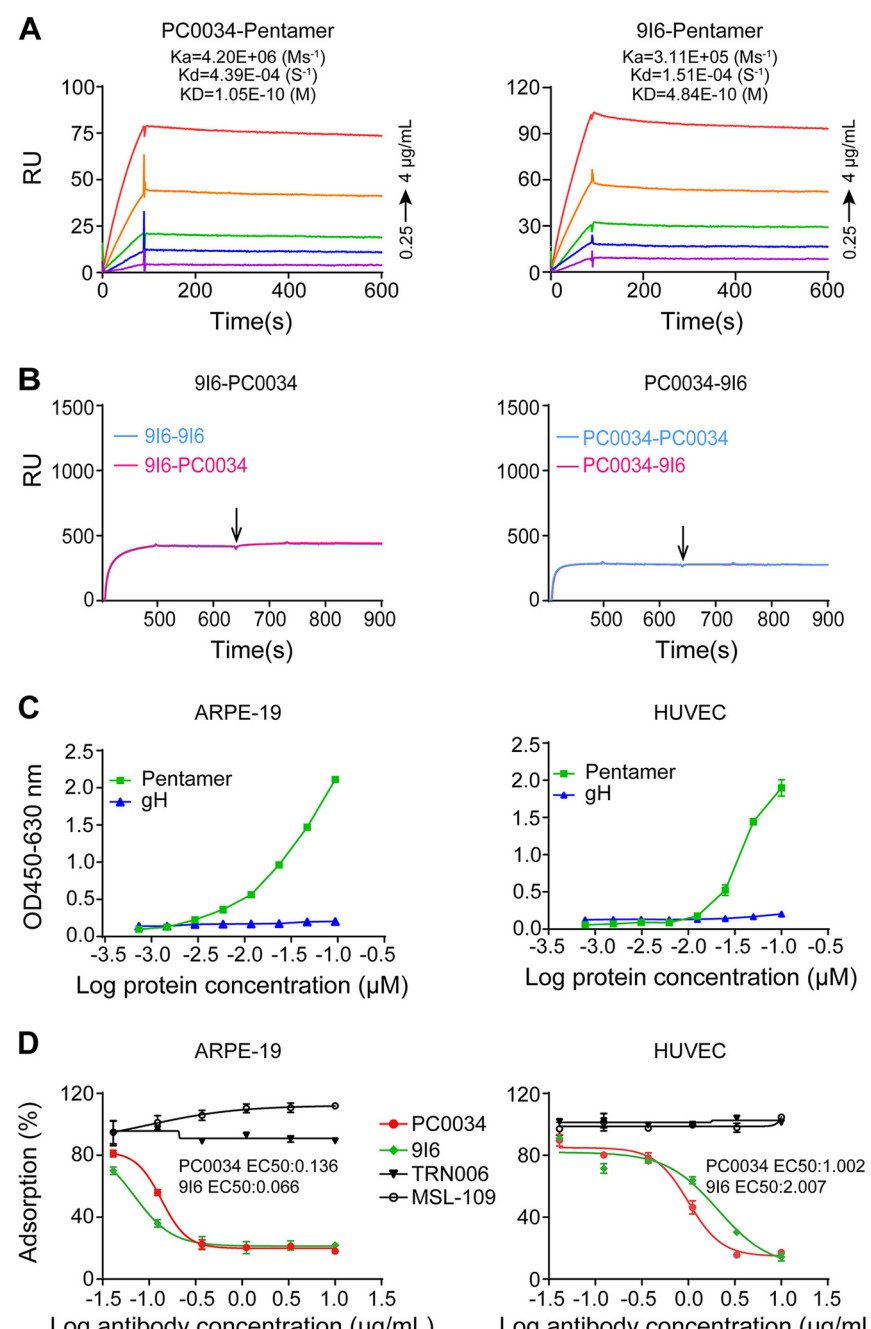

**FIG 8** Binding affinity, cross-reactivity, and inhibition of the binding of pentamer as a surrogate for CMV virions to epithelial/endothelial cells by MAb PC0034. (A) Binding affinities of MAb PC0034 to pentamer in comparison with the known pentamer-specific MAb 9I6 were measured by SPR. (B) The ability of PC0034 and 9I6 to cross-block each other to bind CMV pentamer was determined by SPR similarly to that shown in Fig. 5C. (C) The ability of pentamer as a surrogate for CMV virions at eight 2-fold serial dilutions at the starting concentration of 0.1 $\mu$M to bind to ARPE-19 and HUVEC cells was examined by cell ELISAs (CELISAs). Recombinant gH protein was used as a negative control. (D) The ability of the indicated antibodies in six serial 3-fold dilutions at the starting concentration of 10 $\mu$g/mL to inhibit the binding of pentamer to ARPE-19 cell and HUVEC surfaces was also evaluated in CELISAs. MAb 9I6 was used as a positive control. MSL-109 and TRN006 were used as negative controls. Shown are percentages of the change in pentamer binding affected by the tested antibodies in comparison with the binding measured in the pentamer-only group. The binding of pentamer to both ARPE-19 cells and HUVECs was inhibited by PC0034 and 9I6 with EC$_{50}$ values as indicated but not by MSL-109 or TRN006.

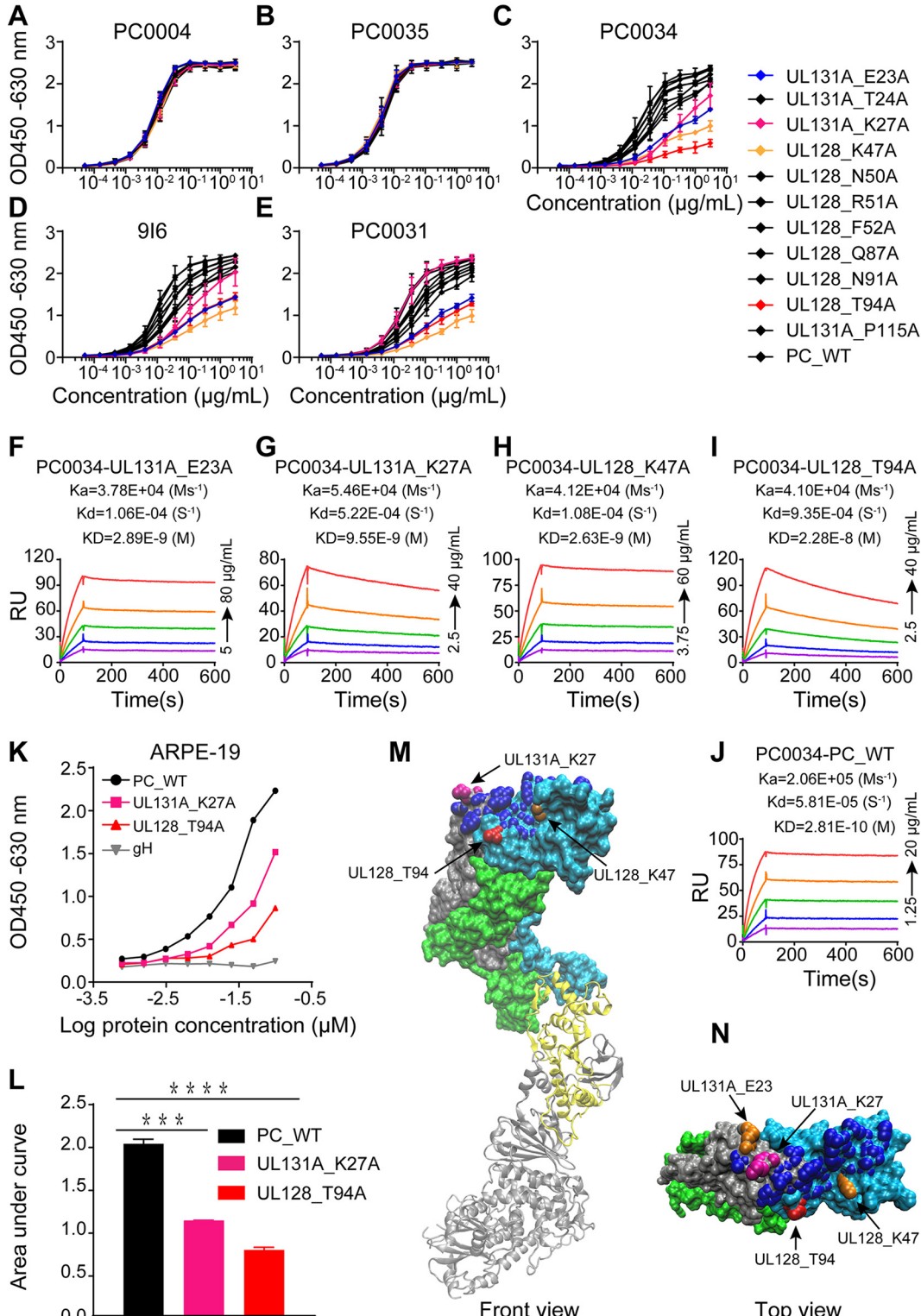

**FIG 9** Epitope mapping of antibody PC0034. (A to E) A panel of 11 pentamer mutants with a single amino acid mutation in subunit UL128 or UL131 using the wild-type pentamer (PC_WT) as a control were evaluated for binding by PC0004, PC0035, PC0034, 9I6, and PC0031 in ELISAs. Pentamer mutants with significantly decreased binding by MAb PC0034 are shown in colors for visibility. (F to J) Binding affinity of four pentamer mutants, UL131A_E23A, UL131A_K27A, UL128_K47A, and UL128_T94A, with PC0034 was determined and compared with that of PC_WT by SPR. Kinetic rate constants and $K_D$ values are shown in the individual plots. (K) Two pentamer mutants, UL128_T94A with approximately 81-fold lower binding affinity and UL131A_K27A with approximately 34-fold lower binding affinity, were tested for binding to ARPE-19 cells in a CELISA. PC_WT and recombinant gH were used as positive and negative controls, respectively. (L) The areas

**TABLE 3** Binding affinity of pentamer-specific antibodies to CMV wild-type pentamer and pentamer mutants[a]

| Mutant | Antibody | $K_a \times 10^4$ (M$^{-1}$ s$^{-1}$) | $K_d \times 10^{-4}$ (s$^{-1}$) | $K_D$ (nM) | Fold change |
|---|---|---|---|---|---|
| UL131A_E23A | PC0031 | 3.47 | 1.64 | 4.80 | 4.36 |
| | PC0034 | 3.78 | 1.06 | 2.80 | 9.96 |
| | 9I6 | 4.07 | 1.15 | 2.83 | 3.43 |
| UL131A_K27A | PC0031 | 11.3 | 0.62 | 0.55 | 0.5 |
| | PC0034 | 5.46 | **5.22** | 9.55 | **33.93** |
| | 9I6 | 9.47 | 1.81 | 1.91 | 2.32 |
| UL128_K47A | PC0031 | 3.43 | 1.42 | 4.15 | 3.76 |
| | PC0034 | 4.12 | 1.08 | 2.63 | 9.33 |
| | 9I6 | 4.92 | 0.738 | 1.50 | 1.82 |
| UL128_T94A | PC0031 | 3.46 | 0.27 | 0.79 | 0.71 |
| | PC0034 | 4.1 | **9.35** | 22.8 | **81.06** |
| | 9I6 | 5.31 | 0.3 | 0.57 | 0.69 |
| PC_WT | PC0031 | 10.5 | 1.16 | 1.1 | |
| | PC0034 | 20.6 | 0.58 | 0.28 | |
| | 9I6 | 16.8 | 1.39 | 0.82 | |

[a]Boldface values indicate the fold decrease in affinity, the ratio of the affinity constant of the pentameric mutants to the affinity constant of the wild-type pentamer.

and B), whereas the binding of PC0034 to the pentameric mutants UL131A_E23A, UL131A_K27A, UL128_K47A, and UL128_T94A was substantially decreased (Fig. 9C). These results indicate that the decrease in binding of PC0034 to these four pentameric mutants indeed resulted from the change in amino acid residues at these positions and was not due to variation in quality of the recombinant pentamer variants produced. Similar decreases in binding of MAb 9I6 to pentameric mutants with these four mutations were detected in ELISAs (Fig. 9D), while binding of PC0031 to pentamer could be affected only by three (UL131A_E23A, UL128_K47A, and UL128_T94A) of these four sites (Fig. 9E). The binding affinity of PC0034 to these four variants assessed by SPR was decreased by 9.96-fold for UL131A_E23A, 33.93-fold for UL131A_K27A, 9.33-fold for UL128_K47A, and 81.06-fold for UL128_T94A (Fig. 9F to I and Table 3). However, the decrease in affinity of MAbs 9I6 and PC0031 to these four variants was much less, as detected by SPR (Table 3). In spite of the fact that two mutants, UL128_T94A and UL131A_K27A, with ~81-fold and ~34-fold decreases in affinity, respectively, had the strongest impact on the binding of PC0034 to the pentamer, these two mutants had very little impact (~0.7-fold and ~2.3-fold) on the binding affinity to MAb 9I6. These results indicate that the PC0034 binding epitope may be slightly different from that of 9I6.

To test whether variants UL128_T94A and UL131A_K27A, with the strongest effect on the binding of pentamer to PC0034, would also affect the binding of these pentameric mutants to host cells, we tested the binding of these two variants to epithelial cells in CELISA. We found that the binding of these two pentameric variants to epithelial cells was also markedly decreased in comparison to that of the wild-type pentamer (Fig. 9K and L). In spite of the fact that these two critical amino acid residues are located in different subunits in the pentamer, structural simulation analysis revealed

**FIG 9** Legend (Continued)
under the curve (AUC) of the wild-type and the two pentamer mutants in panel K were calculated and compared by a statistical two-tailed unpaired $t$ test. $P$ values of $\leq$0.001 (***) and $\leq$0.0001 (****) between the compared groups are indicated. (M and N) Structural features of the front and top views of the PC0034 epitope on UL131 and UL128 subunits in pentamer complex displayed by Visual Molecular Dynamics (VMD). Residues of subunits in pentamer are labeled using the following color scheme: silver for gH, yellow for gL, blue for UL128, green for UL130, and gray for UL131A. Amino acid residues in dark blue are the reported epitopes of 9I6, the amino acid residue in red of UL128_T94 is the unique amino acid residue targeted by PC0034, and the pink amino acid residue in UL131A_K27 and the orange amino acid residue in UL128_K47 and UL131A_E23 might be targeted by both 9I6 and PC0034.

that residues UL128_T94 and UL131A_K27 are located at the close proximity of a distance of 17.15 Å from each other in the pentamer complex, with sufficient exposure on the surface (Fig. 9M and N). These results suggest that the antigen epitope recognized by MAb PC0034 is involved in a conformational patch on the pentamer complex. Finally, sequence analysis of UL128 and UL131 subunits from a total of 214 HCMV genome sequences currently in the NCBI database showed that amino acids UL128_T94 and UL131A_K27 were completely conserved, with no variants found (data not shown).

## DISCUSSION

CMV pentamer plays a pivotal role in viral entry into nonfibroblasts. CMV pentamer is one of the main targets for neutralizing antibodies and has gradually become an integral part in the development of subunit vaccines (49). Pentamer was discovered later than the other important virus entry-related HCMV surface glycoproteins and complexes, i.e., gB, gH/gL/gO, and gM/gN (50). It has been reported that the abundance of trimer (gH/gL/gO) on the surface of HCMV virus is much greater than that of pentamer (51), and both trimer and pentamer contain gH/gL epitopes. Therefore, the total content of the pUL128/UL130/UL131A epitope on the virus surface is lower than that of gH/gL protein. Although the number of antibodies isolated in this study is limited, the number of pentamer-specific antibodies derived from two different individuals was less than that of gH/gL-reactive antibodies. These results might reflect the host immune responses to the abundance of trimer gH/gL epitope present on the CMV of HCMV virions. Since pentamer-specific antibodies play a positive role in the prevention of mother-to-child transmission of CMV and the protection of solid-organ transplant recipients (33, 34), pentamer as a vaccine immunogen has been regarded as an important area in the optimization of CMV vaccines.

Since an effective HCMV vaccine is currently not yet available, expanded solutions for prevention and treatment are needed to more adequately control HCMV infection in immunocompromised individuals. Antibodies targeting HCMV trimer and pentamer are effective in blocking HCMV infection of different cell types. Therefore, targeting of a conserved region and a functionally relevant gH protein has been considered a strategy for induction on broadly neutralizing antibodies (35). In this study, we isolated two groups of MAbs. Antibodies in group 1 target gH/gL in trimeric and pentameric complexes. Three of six antibodies (PC0012, PC0014 and PC0035) in group 1 are neutralizing and compete with the known gH/gL-neutralizing antibody MSL-109 on pentamer. MSL-109 was developed as a therapeutic against HCMV infection and failed in phase II clinical trials. One possible reason for the failure was the development of nongenetic MSL-109 virus-resistant mutants due to the inability of MSL-109 to bind cell-free CMV virions (52). Among the MSL-109-resistant mutants, W168C/R, P171H/S, and D446N mutations in gH contributed to the neutralization resistance to MSL-109. Further studies showed that gH/gL with W168C/R mutations could not be recognized by MSL-109, while gH/gL with P171 (P171H/S) and D446 mutations could still bind MSL-109 (38). In spite of the fact that MAbs PC0012, PC0014, and PC0035 cross-block the binding of MSL-109 to gH/gL to various degrees, they all bind strongly to pentamer mutants (W168A, P171A, and D446A) (Fig. 7A), while MAb MSL-109 did not bind or bound weakly to these mutants. These results suggest that these three newly isolated neutralizing MAbs recognize an epitope(s) that is different from, but partially overlapping or near to, the epitope recognized by MSL-109.

We demonstrated that MAbs PC0012, PC0014, and PC0035 neutralize HCMV by a mechanism that does not block the binding of HCMV virions to cells but rather is involved in the interference of the postattachment step of the virus entry into the host cells.

It has been previously reported that nonneutralizing antibodies to HCMV gB protein can provide protection through antibody-dependent cell-mediated cytotoxicity (ADCC) against HCMV infections (53, 54). There is no report regarding the protective role of

nonneutralizing pentamer-reactive antibodies. We tested the ADCC activity of the HCMV Towne nonneutralizing MAbs PC0004, PC0010, and PC0037 in group 1 (data not shown) but did not identify detectable ADCC activity for these antibodies. The other effector functions, such as antibody-dependent cellular phagocytosis (ADCP) and complement-dependent cytotoxicity (CDC), of MAbs PC004, PC0010, and PC0037 remain to be proven. In addition, since gH is known to induce strain-specific neutralizing antibody responses (55–57), in spite of the fact that MAbs PC0004, PC0010, and PC0037 did not neutralize the HCMV Towne strain and have not been tested for the ability to neutralize strain VR1814 in epithelial and endothelial cells or to bind VR1814-infected cells, the possibility that these three antibodies were capable of neutralizing HCMV in epithelial and endothelial cells could not be ruled out.

It is worth noting that neutralizing MAbs PC0012, PC0014, and PC0035 bind to trimer with much higher affinity than they do to pentamer, suggesting that the gH/gL epitopes on trimer and pentamer are slightly different, but the epitope on trimer may fit better than that on pentamer with these MAbs. It is conceivable that these antibodies were derived from B cells that were initially activated by and matured through the stimulation of the gH/gL epitope on trimer. Conversely, PC0004 bound trimer approximately 20-fold less efficiently than it bound pentamer. It is highly likely that PC0004 antibody was derived from a B cell that was initially activated by and matured through the stimulation of the gH/gL epitope on pentamer. The significance of the difference remains to be elucidated.

The current understanding of the function of pentamer involves the entry of CMV into epithelial cells, endothelial cells, and bone marrow cells by the interaction of pentamer with cell surface receptors, resulting in pentamer-mediated endocytosis (17, 58, 59). Several surface receptors that may interact with pentamer have been identified. Martinez-Martin et al. confirmed that neuropilin 2 (Nrp2) is a receptor for pentamer entry into epithelial and endothelial cells (45). We have investigated the interaction of pentamer with the relevant receptors by recombinant soluble Nrp2 (M1-P864) with His tag and by Nrp2 (M1-P864) with Fc fusion protein. Unfortunately, we did not detect the binding of Nrp2 to pentamer using the Biacore X100 system, and neither was subjected to further analyses. Further, CD147 and OR14I1, members of the olfactory receptor family, are also necessary for pentamer-dependent entry into epithelial cells (60, 61). However, CD147 does not directly interact with the pentamer (60). Cell membrane receptor CD46 may also be involved in the CD46-dependent entry pathway of virus into epithelial cells during congenital infection, and CD46 may interact weakly with pentamer (62). Subsequently, adipocyte membrane-associated proteins (APMAP) were identified and may play a regulatory role in the early stages of HCMV infection by interacting with glycoprotein complex containing gH/gL at a low pH or by mediating nuclear translocation of PP65 (63), but the specific mechanism of APMAP in HCMV infection and whether it interacts with pentamer need further studies for confirmation. Therefore, the discovery of receptors that interact with pentamer and the uncovering of key epitopes for pentamer-cell binding may also provide clues for therapeutic discovery and better vaccine design.

Based on the spatial structure of pentamer, Ciferri et al. classified the currently known pentamer-specific neutralizing antibodies according to the binding region on pentamer and cross-competition into seven different binding sites; of these, MAb 9I6 is located at site 5 (44). The work of Chandramouli et al. resolved the spatial structure of the 9I6 Fab-pentamer complex at a resolution of 5.9 Å and confirmed that the epitope of 9I6 is located in the UL128 and UL131A subunit regions at the top of the pentamer (47). Then, the work of Martinez-Martin et al. elucidated that neutralizing antibodies that bind site 5 strongly block the binding of pentamer to the cell surface receptor Nrp2 (45). Thus, these reported studies indicate that the epitopes on UL128/UL130/UL131A play critical roles in the binding of pentamer to the cell receptors and that neutralizing antibodies that primarily recognize site 5 block the binding of pentamer to cell surface receptors. In the present study, we demonstrated that MAb PC0034 in group 2 recognizes an epitope present in pentamer, cross-blocks the binding of MAb 9I6 to pentamer, and neutralizes HCMV wild-type strain in epithelial cells but does not

neutralize HCMV laboratory standard strain Towne in fibroblast cells. We further showed that MAb PC0034 neutralizes HCMV likely by a mechanism in which PC0034 blocks the attachment of virus to host cells, since PC0034 has been demonstrated to block the adsorption of pentamer to epithelial/endothelial cells in CELISAs. These results suggest that PC0034 most likely binds an epitope at site 5.

Previous studies have shown that mutations at the UL131A_K27 residue potentially affected pentamer binding to Nrp2 (45). Epitope analysis in our study showed that residues UL128_T94 and UL131A_K27 are the key amino acid residues recognized by PC0034 on the epitope. However, UL131A_K27, one of two critically important residues identified for PC0034, is not required for binding by MAb 9I6. It has been reported that epitopes of MAb 9I6 are especially distributed on the UL128 and UL131A subunits at the tip of pentamer, so that single amino acid mutations such as UL128_K47, UL131A_E23, and UL131A_K27 on the epitopes of 9I6 might have little effect on the binding of recombinant pentamer to 9I6. In contrast, amino acid residues UL131A_K27 and UL128_T94 targeted by PC0034 are fully exposed and located in a mountain-like "ridge" position composed of UL128 and UL131A subunits (Fig. 9N). Thus, PC0034 epitope may be a much more exposed target for the induction of antibody response, and any change on the epitope could easily affect the binding of antibody such as PC0034 to pentamer. Nonetheless, this speculation should be confirmed by more precise epitope structure analysis by crystallographic X-ray diffraction and/or cryo-electron microscopy of the PC0034 and CMV pentamer complex.

## MATERIALS AND METHODS

**Cells and viruses.** The ARPE-19 cell line derived from human retinal pigmented epithelial cells, MRC-5 cells derived from human lung fibroblasts, and cells derived from primary human umbilical vein endothelial cells (HUVECs) were all purchased from ATCC through BNCC (Chinese distributor for ATCC). The laboratory standard CMV Towne strain expressing green fluorescent protein (GFP) as the reporter was kindly provided by Hua Zhu, HCMV strains VR1814, NR, and AD169 FIX were kindly provided by Hua Zhu, and HCMV strain BE13/2012 was kindly provided by Mingli Wang.

**Production of recombinant HCMV glycoproteins and variants.** The intronless full-length DNA sequence encoding gL, gO, UL128, UL130, and UL131A of HCMV strain VR1814 (accession number GU179289.1) was obtained from GenBank, codon optimized for expression in human mammalian cells, *de novo* synthesized, and subcloned into the eukaryotic plasmid expression vector pcDNA3.1$^+$ (Invitrogen) by GenScript. To produce soluble pentamer and trimer and facilitate purification, the designed gH gene expression construct contained the sequence encoding the extracellular region (residues 1 to 715) of gH protein and the sequence encoding a C-terminal 6×His tag. For epitope mapping, 11 pentamer mutants with single amino acid substitutions of alanine at seven different positions in subunit UL128 and at four different positions in subunit UL131 (Fig. 9) and three gH mutants with a single amino acid substitution of alanine were produced by use of a Q5 site-directed mutagenesis kit (NEB) using the wild-type pentamer and gH expression constructs as the templates. Recombinant gH/gL/UL128/UL130/UL131A pentamer was produced in HEK293F by cotransfection with gH, gL, UL128, UL130, and UL131A plasmids with a mass ratio of 1:0.8:0.6:0.6:0.6 by use of a previously described method (33), while the gH/gL/gO trimer was produced in HEK293F by cotransfection with gH, gL, and gO plasmids at a mass ratio of 1:1:1. Then, recombinant HCMV glycoproteins were purified by Ni-NTA agarose chromatography (GE Healthcare) and analyzed by using 4% to 12% bis-Tris SDS-PAGE gels (Thermo Fisher). Purified pentamer and trimer complexes were subjected to further purification and analysis by size exclusion chromatography (SEC) using an Agilent 1260 system (Agilent Technologies). The purified proteins were stored at −80°C before use.

**PBMCs and sorting of specific single B cells by fluorescence-activated cell sorting (FACS).** Peripheral blood samples were obtained from HCMV-positive healthy volunteers in accordance with the Institutional Review Board protocols approved by the Jinan University Hospital Institutional Review Board. Plasma and PBMCs were isolated from blood using Ficoll-Paque PLUS (GE Healthcare; catalog no. 17-1440-02) density gradient centrifugation. Plasma samples were used for screening samples with high antibody titers to HCMV pentamer. PBMCs from individuals with high antibody titers to pentamer were used for sorting single B cells.

HCMV pentamer-reactive single memory B cells were identified by staining with fluorescence color-labeled antibodies against the surface markers of CD3-phycoerythrin (PE)-Cy5, CD14-fluorescein isothiocyanate (FITC), CD16-PE-Cy5, CD235a-PE-Cy5, CD20-allophycocyanin (APC), CD27-APC-H7, IgD-PE, and recombinant pentamer probe labeled separately with fluorescence dye BV421 and PE-Cy7 through biotin-streptavidin conjugation (BioLegend) by a method described previously (64). All antibodies were titrated and used at optimal concentrations determined in advance. Single B cells with CD3$^-$, CD14$^-$, CD16$^-$, CD235a$^-$, IgD$^-$, CD20$^+$, and CD27$^+$, and dual color pentamer-positive surface markers were sorted by a BD FACS Aria III into microtiter 96-well PCR plates containing cell lysis reverse transcription buffer for immunoglobulin (Ig) gene amplification. Flow cytometry data were analyzed using Diva software.

**Isolation of human pentamer-reactive antibody $V_H D_H J_H$ and $V_L J_L$ genes.** Ig variable regions of heavy-chain ($V_H D_H J_H$) and light-chain ($V_L J_L$) genes were amplified by reverse transcription-PCR from the sorted pentamer-specific single B cells. PCR products from each individual single B cell were sequenced and used for assembling linear full-length IgG1 and light-chain gene expression cassettes for production of recombinant antibodies for screening using methods described previously (37, 65). Ig $V_H D_H J_H$ and $V_L J_L$ gene sequences were annotated for gene family, clonal lineage, somatic mutation frequency, and CDR3 length using the IMGT information system (http://www.imgt.org).

**Expression and purification of IgG1 recombinant antibody.** For initial screening to identify pentamer-reactive antibodies, recombinant antibodies were produced in 293T cells by transient transfection with the purified naturally paired linear full-length IgG1 and light-chain gene expression cassettes by a method described previously (65). $V_H D J_H$ and $V_L J_L$ genes of the full-length IgG1 and light-chain genes of the eight representative pentamer-reactive antibodies (see boldface in Table 1) were subcloned into pCDNA3.1$^+$ (Invitrogen) mammalian expression vector for production antibodies for further characterization. Used as references and for comparative study, gene expression constructs based on the amino acid sequences of MAb MSL-109, known as gH/gL-reactive neutralizing antibody, and MAbs 8I21 and 9I6, known as pentamer-specific neutralizing antibodies, obtained from the Protein Data Bank (PDB ID 4LRI, 5VOB, and 5VOD, respectively) were derived, codon optimized, de novo synthesized, and cloned into the pCDNA3.1$^+$ (Invitrogen) vector. Recombinant antibodies were produced in 293F cells by transient transfection and purified from supernatants of the transfected cell cultures using protein A columns by a method described previously (65).

**Binding of MAbs with HCMV pentamer and trimer in ELISAs.** The ability of purified MAbs to bind HCMV gH/gL/UL128/UL130/UL131A pentamer and gH/gL/gO trimer were measured in ELISAs. F96 MaxiSorp Nunc-IMMUNO 96-well plates (Thermal Fisher scientific, Roskilde, Denmark) were coated with either HCMV pentamer or trimer at 100 ng/well in carbonate bicarbonate buffer at pH 7.4 and incubated overnight at 4°C. Plates were washed twice with PBST (phosphate-buffered saline [PBS]–0.1% Tween 20) and blocked with PBS containing 5% goat serum for 1 h at room temperature (RT). Plates were then washed three times with PBST. The tested antibodies, in eight serial 3-fold dilutions at the starting concentration of 1 $\mu$g/mL (100 ng/well), were added to the plates and incubated at 37°C for 1 h. An antigen-irrelevant human IgG antibody, TRN006, was used as a negative control. HCMV-neutralizing antibodies MSL-109, 8I21, and 9I6 were used as positive controls. After washing three times with PBST, goat anti-human IgG (H+L)–horseradish peroxidase (HRP) (Promega) at a 1:10,000 dilution was added at 100 $\mu$L/well and incubated at 37°C for 1 h. After washing with PBST, HRP substrate TMB, 3,3',5,5'-tetramethylbenzidine (Solarbio Life Sciences) was added at 100 $\mu$L/well and incubated at 37°C for 5 min. A stop solution of 2 M $H_2SO_4$ was added at 50 $\mu$L/well to stop the reaction. Reaction plates were read at 450/630 nm in a SpectraMax Paradigm microplate reader (Molecular Devices).

**HCMV neutralization assays.** For HCMV Towne strain neutralization assays, MRC-5 cells were seeded at $4 \times 10^4$ cells per well in 96-well plates (Corning, Inc., USA). Aliquots of 50 $\mu$L/well of HCMV Towne at a multiplicity of infection (MOI) of 1 were mixed with 10 $\mu$L/well of antibodies (60 $\mu$g/mL) in eight serial 3-fold dilutions at the starting antibody concentration of 10 $\mu$g/mL and were incubated at 37°C for 1 h. After incubation, the mixtures of 60 $\mu$L/well were added to MRC-5 cells in 96-well plates and incubated for 4 h, followed by the addition of 100 $\mu$L/well of culture medium containing 2% fetal bovine serum (FBS), and then cultured at 37°C in a 5% $CO_2$ incubator for 5 days with a change of the culture medium once on the third day. After removal of the culture medium, plates were fixed with 4% paraformaldehyde for 30 min at RT, washed twice with PBST, and then treated with 0.1% Triton X-100 (Sigma) for 10 min, washed twice again with PBST, and blocked with PBS containing 5% goat serum for 1 h at RT. The infectivity of the plates was detected by incubation with 1 mg/mL (100 $\mu$L/well) of known anti-HCMV gB antibody 8F9 (66) for 2 h at 37°C. After washing three times with PBST (PBS–0.1% Tween 20), goat anti-human IgG (H+L)–HRP (1:10,000 dilution; Promega) was added at 100 $\mu$L/well and incubated at 37°C for 1 h. After washing with PBST, HRP substrate TMB (Solarbio Life Sciences) was added at 100 $\mu$L/well and incubated at 37°C for 5 min. Then, 2 M $H_2SO_4$ solution was added to stop the reaction. Plates were read at 450/630 nm in a SpectraMax Paradigm reader (Molecular Devices).

Neutralization assays for the clinical isolate BE13/2012 in MRC-5 cells were evaluated by counting HCMV-specific cytopathic plaques. MRC-5 cells in 96-well plates were infected with approximately 200 HCMV infectious virions per well premixed with the tested antibodies in eight serial 3-fold dilutions at the starting concentration of 10 $\mu$g/mL. The infected cultures were incubated at 37°C, in 5% $CO_2$, for 48 h, fixed in 4% paraformaldehyde for 30 min, washed twice with PBST, treated with 0.1% Triton X-100 (Sigma) for 10 min, followed by two more washes with PBST, and then blocked with PBS–5% goat serum for 1 h at RT. The plates were incubated with mouse anti-HCMV IE1/IE2 antibody (1:500 dilution) (Abcam) for 2 h at RT. The plates were washed three times with PBST and incubated with HRP-conjugated goat anti-mouse IgG (1:10,000 dilution) at RT for 2 h. After washing four times with PBST, 50 $\mu$L/well of DAB, 3,3'-diaminobenzidine (MXB Biotechnologies, China) was added and the plates were incubated for 6 min at RT, followed by the addition of 100 $\mu$L/well of double-distilled water (ddH$_2$O) to stop the reaction. Images of the whole wells were taken using an EVOS FL Auto 2 (Thermo Fisher) microscope. HCMV-specific cytopathic plaques were counted by using ImageJ software.

Neutralization assays for HCMV strains VR1814, NR, and AD169 FIX in ARPE-19 cells were carried out similarly to the method used for assaying BE13/2012. Briefly, aliquots of approximately 500 virions of HCMV strain VR1814, NR, or AD169 FIX were premixed with antibodies in eight serial 3-fold dilutions at the starting antibody concentration of 1.6 $\mu$g/mL and incubated for 1 h at 37°C. The mixtures were added to confluent monolayers of ARPE-19 cells in 96-well plates and incubated at 37°C, in 5% $CO_2$, for

48 h. The infectivity of the plates was detected by a measurement method similar to that for BE12/2013 described above, while the secondary antibody was goat anti-mouse IgG conjugated with FITC.

**Cross-competitive blocking competition binding ELISA.** Aliquots of the tested MAbs were biotinylated using EZ-Link sulfo-NHS-biotin (Thermo Scientific) in accordance with the manufacturer's protocol. Working concentrations of the individual biotinylated antibodies (detection) were determined by titration to be near saturation based on the linear range of the corresponding binding values. F96 MaxiSorp Nunc-IMMUNO 96-well plates (Thermal Fisher scientific, Roskilde, Denmark) were coated with pentamer protein at 100 ng/well in carbonate bicarbonate buffer at pH 7.4 and incubated overnight at 4°C. Plates were blocked with PBS containing 0.1% Tween 20 and 5% (wt/vol) goat serum at RT for 2 h. Each of the tested antibodies was assayed for cross-blocking each other by a checkerboard arrangement in ELISAs. Aliquots of blocking antibodies in eight serial 3-fold dilutions starting at the concentration of 10 $\mu$g/mL were added to the 96-well plates. Simultaneously, the biotin-labeled antibodies at the predetermined subsaturated concentration were added and incubated at 37°C for 1 h, followed by washing three times with PBST. Streptavidin-HRP (Abcam) at 100 ng/well was then added and incubated at 37°C for 1 h. After washing with PBST, HRP substrate TMB (Solarbio Life Sciences) was added at 100 $\mu$L/well and incubated at 37°C for 5 min. A 2 M $H_2SO_4$ solution was added to stop the reaction. Plates were read at 450/630 nm in a SpectraMax Paradigm reader (Molecular Devices). The percent change in binding of antibodies to pentamer was determined by comparing values of optical density at 450 nm ($OD_{450}$) of the biotin-labeled antibodies in the presence or absence of the unlabeled blocking antibodies.

**Binding kinetics and competitive binding of antibodies to pentamer and trimer in SPR assays.** The dissociation rate constant ($K_d$), association rate constant ($K_a$), and affinity constant ($K_D$) values of antibodies to trimer, pentamer, and pentamer variants were measured by SPR on a Biacore X100 instrument (GE Healthcare). First, anti-human Fc IgG antibody was immobilized on channel 2 of a CM5 chip by amine coupling using a human antibody capture kit (GE Healthcare). The tested antibodies at 4 $\mu$g/mL were captured by the anti-human Fc IgG antibody on a CM5 chip to approximately 100 response units (RU). Channel 1 of the CM5 chip served as the background reference channel. Recombinant HCMV pentamer or trimer in five 2-fold serial dilutions were injected and allowed to flow over the antibodies captured on the chip at a rate of 30 $\mu$L/min for 90 s with a 600-s dissociation time. The chip was regenerated by injection of 3 M $MgCl_2$ for 30 s. Curves were fitted to a 1:1 binding model to determine kinetic rate constants ($K_a$ and $K_d$). $K_D$ values were calculated from these rate constants.

In antibody binding competition experiments, anti-His tag antibody was immobilized on a CM5 chip by amine coupling using a His capture kit (GE Healthcare) resulting in 10,211 RU coupled on channel 1 and 10,206.3 RU coupled on channel 2. Channel 1 served as a reference channel. Purified pentamer with 6×His tag at a concentration of 20 $\mu$g/mL was injected into channel 2 at a rate of 5 $\mu$L/min for 180 s. The first antibodies at a concentration of 200 $\mu$g/mL were injected and allowed to flow at a rate of 10 $\mu$L/min for 90 s, followed by injection of the secondary antibodies at a concentration of 200 $\mu$g/mL at a flow rate of 10 $\mu$L/min for 90 s. The chip was regenerated by injection of glycine (pH 1.5) for 30 s. All experiments were performed at RT, and data analyses were carried out using Biacore X100 evaluation software (version 2.0.1).

**HCMV adsorption and postattachment neutralization assays.** In virus adsorption assays, MRC-5 cells at a density of $8 \times 10^4$ were seeded per well of 24-well plates (Corning, Inc., USA). Individual MAbs at a concentration of 10 $\mu$g/mL were mixed with HCMV Towne strain at an MOI of 0.5, incubated at 37°C for 1 h, and then cooled at 4°C for 15 min. The cooled antibody-virus mixtures were then added to MRC-5 cells in 24-well plates, also precooled at 4°C for 15 min, and incubated at 4°C for 1 h to allow virus to bind to cells. Virus DNA isolation was performed at 1 h postinfection. Heparin was demonstrated to block adsorption of CMV to cells (41) and was used as a positive control. Cells in the plates were washed three times with PBS chilled to 4°C and lysed with lysis buffer. DNAs were extracted from cell lysates using an E.Z.N.A. viral DNA kit (Omega). DNA copy numbers of the HCMV-specific gene *UL75* were determined by real-time quantitative PCR using sets of primers specific for *GAPDH* as the housekeeping gene for normalization (*qGAPDH*_F, AGAAGGCTGGGGCTCATTTG; *qGAPDH*_R, AGGGGCCATCCACAGTCTTC) and primers specific for *qUL75*_F (GGTTTCTAGCGTGCTTGGTTGCATG) and *qUL75*_R (CAGCGACCTGTACACACCCTGTTCC).

In the postattachment neutralization assays, MRC-5 cells were seeded at $4 \times 10^4$ cells per well in 96-well plates (Corning, Inc., USA). The precooled cells were inoculated with precooled Towne strain virus at an MOI of 1 and incubated at 4°C for 30 min. The virus inoculum was removed and washed once with ice-cold PBS. Antibodies at a concentration of 10, 1, or 0.1 $\mu$g/mL in culture medium were then added to the cells and incubated at 37°C, in 5% $CO_2$, for 96 h. After removal of the culture contents, the plates were fixed with 4% paraformaldehyde for 30 min, followed by washing twice with PBS. The infectivity of the infected cultures was determined by the intracellular fluorescence intensity of GFP using a SpectraMax Paradigm reader (Molecular Devices).

**HCMV cell-to-cell spread assay.** For the HCMV Towne strain spread assay, $4 \times 10^4$ MRC-5 cells grown in 96-well plates (Corning, Inc., USA) were infected with aliquots of 50 $\mu$L/well of HCMV Towne at about 200 PFU. MSL-109 and an anti-gB antigenic domain 5-neutralizing MAb ($\alpha$-gB), which was isolated from an HCMV serum-positive individual and a gift from Trinomab, were used as a positive control. MAb TRN006 with irrelevant antigenic specificity was used as a negative control. At 4 h postinfection, aliquots of 50 $\mu$L/well of fresh medium containing 100 $\mu$g/mL of either MSL-109, PC0012, PC0014, PC0035, TRN006, or $\alpha$-gB (positive-control IgG) or no antibody was used to replace the culture medium of the infected MRC-5 cells. The infected cultures were incubated at 37°C, in 5% $CO_2$, for 5 days, fixed in 4% paraformaldehyde for 25 min, washed twice with PBST, treated with 0.5% Triton X-100 (Sigma) for 5 min, followed by two washes with PBST, and then blocked with PBS–10% goat serum for 1 h at RT. The plates were incubated with mouse anti-HCMV IE1/IE2 antibody (1:1,000 dilution) (Abcam) for 2 h at RT. The plates were washed three times with PBST and incubated with HRP-conjugated goat anti-mouse IgG (1:10,000 dilution) at RT for 1 h. After washing five times with PBST, 50 $\mu$L/well of DAB (MXB Biotechnologies, China) was added and the plates were incubated for ~2 to 5 min at

**TABLE 4** Accession numbers of the isolated CMV trimer/pentamer-reactive antibodies

| Accession no. | Antibody |
|---|---|
| ON983983 | PC0004 heavy chain |
| ON983984 | PC0004 kappa chain |
| ON983985 | PC0010 heavy chain |
| ON983986 | PC0010 kappa chain |
| ON959468 | PC0012 heavy chain |
| ON959469 | PC0012 kappa chain |
| ON959470 | PC0014 heavy chain |
| ON959471 | PC0014 kappa chain |
| ON983987 | PC0031 heavy chain |
| ON983988 | PC0031 lambda chain |
| ON959472 | PC0034 heavy chain |
| ON959473 | PC0034 lambda chain |
| ON959474 | PC0035 heavy chain |
| ON959475 | PC0035 lambda chain |
| ON983989 | PC0037 heavy chain |
| ON983990 | PC0037 lambda chain |

RT, followed by the addition of 100 $\mu$L/well of ddH$_2$O to stop the reaction. Images were taken using an EVOS FL Auto 2 (Thermo Fisher) microscope.

**Structure preparation and MD simulations.** The binding mechanism of antibody PC0034 to pentamer was explored using a molecular docking (MD) approach. We used the crystal model of antibody 9I6 binding with pentamer (PDB ID 5VOD) as the template and reconstructed the three-dimensional (3D) structural model of PC0034 with pentamer using the Homology module in Discovery Studio v4.5 (DS) software. The best theoretical model was selected from the 20 models by using the probability density function (PDF) total energy and discrete optimized protein energy (DOPE) score (the lower the score, the better the model). Subsequently, the DS_Charmm module was used for energy minimization and molecular dynamics simulations, respectively. The charmm27 force field was used in the Generalized Born with a simple Switching (GBSW) solvent model and calculated long-range electrostatic interactions in the system using the Spherical Cutoff method. Energy minimizations were performed, using the steepest descent and conjugated gradient methods, until convergence to 0.4184 kJ/(mol · nm). The optimized 3D structure was used for epitope analysis, and the antigenic structure in the 5-Å range of the antibody CDR region is the antigenic epitope.

**Binding of pentamer as a surrogate for CMV to epithelial/endothelial cells.** Binding of pentamer to epithelial/endothelial cells was evaluated by a method described previously (47). Briefly, HUVECs and ARPE-19 cells at a density of approximately $4 \times 10^4$ were seeded per well in 96-well plates (Corning, Inc., USA) and cultured in Dulbecco's modified Eagle medium (DMEM; Gibco) containing 10% FBS in a 5% CO$_2$ and 37°C incubator overnight. Aliquots of pentamer or gH (used as a negative control) in eight 2-fold serial dilutions starting at 0.1 $\mu$M with binding medium containing 30 mM HEPES and 5% FBS were added 50 $\mu$L/well to HUVECs and ARPE-19 cells precooled at 4°C for 10 min and incubated for 1 h at 4°C. The cells were washed three times with PBS (Gibco) and fixed with 50 $\mu$L/well of 4% paraformaldehyde for 30 min. After washing three times with PBS, mouse anti-6×His tag–HRP conjugate (Proteintech) at a 1:500 dilution in PBS solution containing 4% FBS was added to 96-well plates and incubated for 1 h at 37°C. Following four washes with PBS, 50 $\mu$L/well of TMB substrate (Solarbio Life Sciences) was added and the plates were incubated at 37°C for 15 min. Then, a 2 M H$_2$SO$_4$ solution was added to stop the reaction. The plates were read at 450/630 nm in a SpectraMax Paradigm reader (Molecular Devices). The higher values of 450/630 absorbance reflect the stronger binding of pentamer protein to HUVECs and ARPE-19 cells.

MAb PC0034 in six serial 3-fold dilutions starting at 10 $\mu$g/mL was assayed for the ability to block the binding of pentamer (0.025 $\mu$M) to HUVECs and ARPE-19 cells with MAb 9I6 as a positive control and antigenic specificity-irrelevant MAb TRN006 and nonblocking CMV-neutralizing antibody MSL-109 as negative controls. Data are presented as the percentages of inhibition of adsorption of pentamer to cells as a function of antibody concentration.

**Data availability.** The heavy- and light-chain variable region sequences of the eight representative pentamer-reactive antibodies were deposited in GenBank under the accession numbers shown in Table 4.

## ACKNOWLEDGMENTS

We thank Mingli Wang for kindly providing virus strains.

This work was supported by the Guangdong Natural Science Foundation Project Fundamental and Applied Fundamental Research Fund (grant no. 2114050001966) and by the Zhuhai Innovative and Entrepreneurial Research Team Program (grant no. ZH01110405160015PWC).

Y.A. and C.W. designed the experiments, performed most of the experiments, and analyzed the data. Y.A., Y.W., T.L., L.Z., and N.L. isolated the antibodies and analyzed related

data. Y.A., D.K.J., H.Z., X.Y., C.L., and M.Z. participated in the discovery and characterization of antibodies. W.Y. completed molecular simulation analysis. Y.A. and W.Z. wrote the original draft. H.-X.L. designed and directed the study, analyzed the data, and supervised the overall work. Y.A., M.Z., and H.-X.L. prepared and edited the manuscript.

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
