## [Reviewer comments · Microbiology Spectrum]

Microbiology Spectrum

Neutralization epitopes in trimer and pentamer complexes recognized by potent CMV neutralizing human monoclonal antibodies

Yuanbao Ai, Changwen Wu, Ming Zhang, Dabbu Jaijyan, Tong Liu, Lipeng Zan, Nan Li, Wei Yu, Yueming Wang, Xiaohui Yuan, Chengming Li, Weihong Zheng, Hua Zhu, and Hua-Xin Liao

Corresponding Author(s): Hua-Xin Liao, Jinan University

Review Timeline:

Submission Date:	April 23, 2022
Editorial Decision:	June 8, 2022
Revision Received:	July 29, 2022
Editorial Decision:	August 13, 2022
Revision Received:	September 20, 2022
Accepted:	September 23, 2022

Editor: Haidong Gu

Reviewer(s): Disclosure of reviewer identity is with reference to reviewer comments included in decision letter(s). The following individuals involved in review of your submission have agreed to reveal their identity: Dai Wang (Reviewer #2); Domenico Tortorella (Reviewer #3)

Transaction Report:

DOI: <https://doi.org/10.1128/spectrum.01393-22>

June 8, 2022

Dr. Yuanbao Ai
Department of Cell Biology, College of Life Science and Technology, Jinan University
Department of Cell Biology
601 Huangpu Avenue West
Guangzhou, GuangDong
China

Re: Spectrum01393-22 (Neutralization epitopes in trimer and pentamer complexes recognized by potent CMV neutralizing human monoclonal antibodies)

Dear Dr. Yuanbao Ai:

Thank you for submitting your manuscript to Microbiology Spectrum. I regret to inform you that your manuscript cannot be accepted for publication in Microbiology Spectrum in the current form. I encourage you to revise the manuscript by: 1) fully addressing the reviewers' concerns, some of which have been raised in previous reviews, and 2) depositing the mAb sequences to GenBank with an embargo until paper is accepted.

Link Not Available

Sincerely,

Haidong Gu

Journals Department
Reviewer comments:

Reviewer #1 (Comments for the Author):

I reviewed an earlier version of this manuscript from Ai and colleagues. Reviewers were impressed by the technical quality of the work, but had concerns about the novelty of results and suggested experiments to address better the mechanism of neutralization. Little to no new experiments were included to address previous reviews. However, changes were made to the writing including more accurate interpretations and presentation of data. Points to consider:

I agree that this paper will be of limited use to the field if sequences are not made public. Data can be deposited with GenBank, and authors can request embargo from GenBank until the paper is accepted.

Since specific receptor interactions are not investigated here, I suggest that the detailed receptor information in the Introduction Lines ~ 103-115 be combined with the Discussion.

Consider mentioning as data not shown that attempts were made to demonstrate receptor interactions with Nrp2 by biosensor.

Reviewer #2 (Comments for the Author):

In this manuscript, Ai et al identified a panel of human monoclonal antibodies against CMV pentameric gH/gL complex from naturally infected donors, and characterized their epitope specificities and abilities for neutralizing CMV infection. Even though no new neutralization mechanism nor binding epitopes were identified, the study does contribute to a growing list of critical resources and tools for the CMV field to better understand the natural CMV immunity, which would potentially inform development of CMV vaccines and therapeutics.

I agree with one of the previous reviewers that the sequences of the antibodies need be disclosed, ideally in the manuscript. I also advise the authors to revise the manuscript to more adequately address the reviewers' comments from the previous submission.

The only additional comment I had is that the authors might want to consider to further characterize the mAbs PC0004, PC0010 and PC0037, and evaluate their ADCC and ADCP activities, or neutralizing potentials in the presence of complement. Any new findings would strengthen the novelty of manuscript. The function of the non-neutralizing antibodies may also play a role in protection. This at least should be discussed.

Reviewer #3 (Comments for the Author):

The manuscript entitled "Neutralization epitopes in trimer and pentamer complexes recognized by potent CMV neutralizing human monoclonal antibodies" by Wu et.al describes the characterization of several monoclonal antibodies that bind and/or neutralize diverse strains of human CMV. The studies utilize molecular binding studies as well as virus neutralizing assays to define the specificity of the identified anti-CMV antibodies. Even though previous studies have identified anti-CMV neutralizing mAbs from human B cells, these studies have provided a detailed molecular analysis of mAb binding and neutralization. Importantly, the studies have defined a neutralizing epitope in the pentamer complex that should be targeted for CMV vaccines.

There are some minor points that should be addressed:

- 1) Lines 31-36 is unclear and difficult to follow. This should be revised.
- 2) Lines 153-157 are awkward and should be revised.
- 3) Line 68 is awkward and should be revised.
- 4) Line 237 is awkward and should be revised.
- 5) Line 386 is is awkward and should be revised.
- 6) The generation of pentamer and trimer should include more detail with the transfection conditions to generate these complexes.

Reviewer #4 (Comments for the Author):

This manuscript by Ai et al. focuses on the natural humoral immune response directed against its trimer or pentamer complexes. They generated a panel of monoclonal antibodies (mAbs) from healthy individuals and characterized 4 neutralizing mAbs from 2 distinct groups. MAbs PC0012, PC0014 and PC0035 in group 1 bind both trimer and pentamer and recognize antigenic epitope(s) clustered in a similar area or region partially overlapped with or close by the epitope recognized by the known neutralizing anti-gH antibody MSL-109. MAb PC0034 from group 2 binds only to pentamer, with binding characteristics similar to the established anti-pentamer antibody 9I6.

Overall, this is a well-conceived study and the data are generally convincing and novel. The results are significant and may contribute to a better understanding of HCMV-entry and how this can be blocked by the humoral immune response against neutralizing epitopes within gH/gL containing complexes.

Minor comments/questions:

This study aimed "to identify key protein(s) and antigenic epitopes in HCMV pentamer complex....for designing better CHMV vaccines.....and provide attractive candidates for the development of an effective cocktail therapeutics" (see lines 52ff). Indeed,

the authors identified novel mAbs against established (or partially overlapping) sites in gH and UL128+UL131. However, the most interesting and novel mAb for my point of view is PC0031, which does not compete with any of the established mAbs like MSL-109 or 9I6 or 8I21 but which was unfortunately not further characterized in this study. Since pentamer is considered as an important vaccine immunogen - characterization of the PC0031 epitope might significantly contribute to the overall aim of this study. Similarly, PC0004 has very distinct binding characteristics in Fig.3 but was not characterized in any of the assays Fig. 5- Fig. 7. The authors should at least provide an explanation for ignoring PC0031 and PC0004.

Lines 21 and 68: consider to use the same numbers for consistency - i.e. "35%" or "36%" .

Line 28: consider to mention where these antibodies were derived from.

Lines 49f: "...no effective HCMV vaccines or HCMV specific therapeutics have been developed". Consider to formulate this more accurately. This is contrary to what the authors state themselves in lines 119f, 125f or 134-139. In addition, there are several HCMV specific therapeutics available (GCV, Letemovir, MBV), however their use is limited by toxicity or drug-resistance.

Line 79: the authors should make clear what is meant with "after the pre-fusion event of the gH/gL complex" - probably "the event" is trimer or pentamer not bound to their receptor(s).

Line 123: "CMVIG exhibit some efficacy against HCMV". The authors may consider to mention recent experience with CMVIG, which showed promising results after SOT (Santhanakrishnan K. et al., 2019; Alsuliman T. et al., 2018; Bonaros N. et al., 2008) or upon biweekly administration efficiently prevented maternal-fetal HCMV transmission after primary infection in the first trimester (Kagan et al., 2019).

Lines 141-143: Consider to mention all the mAbs that belong to either of the two groups. As far as I understood Fig. 3, PC004, PC0010, PC0037 also belong to group 1, while PC0031 additionally belongs to group 2.

Line 153: "studies on the newly identified these two groups of potent" - sounds weird to me.

Line 170: "(PBMC) of four specimens" - Why does Fig. 2B show only two of them - while Fig. 2A shows all four? Fig 2A, shows high titers of pentamer-reactive antibody responses in Es0099 or Es0055, but no monoclonal antibody against trimer or pentamer could be isolated from these donors?

Line 199: do the authors have any idea why PC0004 bound trimer 20fold less efficiently than pentamer (Fig 3)? Given the distinct binding properties of PC0004, wouldn't this mAb be even more important/interesting to include in the assays Fig.4- Fig.7!?

Line 217 and Fig 4B+4C: "...one (PC0034) of them, neutralized HCMV wild strain" - Why do the authors emphasize on this particular mAb? Was PC0031 not tested against all these strains and/or not neutralizing against these strains?

Lines 232ff and Fig. 6: "epitope(s) recognized by mAbs PC0012, PC0014 and PC0035 in comparison with the known neutralizing antibody MSL-109" - ... "and the known anti-pentamer antibodies 9I6 and 8I21".

What about PC0031, which is also included in this figure? Isn't it one of the most intriguing findings of this study that PC0031 neither competes with MSL-109 nor the established anti-pentamer mAbs for binding and therefore targets a novel epitope on UL128-131!?!?

Line 283ff and Fig. 8: Again, I wonder why PC0031 was not included in these assays. Fig. 6 convincingly demonstrated that PC0034 is a 9I6-like antibody and raised the question to what PC0031 binds. This could have been followed up here in Fig. 8.

Line 301: "by blocking the CMV attachment" - consider to change for "binding" to make clear that the mechanism is "post-attachment" as shown in Fig. 7C and written in line 282 of the manuscript.

Line 312: Consider showing the data because PC0004 has drastically reduced trimer binding characteristics in Fig. 3 and a comparable binding of this antibody could serve as a perfect internal control for complex integrity. In addition, rather than using anti-gH antibodies PC0004 and PC0035 for verification of the integrity of the pentamer complex- I would strongly recommend to utilize PC0031, that belongs to group 2 mAbs and more importantly, which according to Fig6 has significantly different binding characteristics and might therefore be valuable to confirm integrity of the UL-proteins in the pentamer-mutated complexes. It should also be clarified in lines 309f, that PC0004 and PC0035 belong to group 1 mAbs (binding to trimer and pentamer) and are therefore gH/gL-specific.

Lines 303: As stated above, Fig. 6 convincingly demonstrated that PC0034 is a 9I6-like antibody. Wouldn't then 9I6 be the perfect control for the experiments with pentamer constructs that harbor mutations in the 9I6 binding site (Fig. 9). Similar to what was done with MSL-109 and its binding to the respective gH mutants in Fig. 7A. This would help to prove or disprove that PC0034 is identical/similar or different to this established mAb 9I6 - and this is one of the main aims of the study!?!?

Line 503 "recombinant antibodies were produced in 293T...." - this sentence should be moved after line 507 "Expression and purification of IgG1 recombinant antibody". Was the antibody produced in 293T as stated here or in 293F as written in lines 509 +521?

Line 512 "16 pentamer-reactive antibodies were cloned....and produced" - this opposes to what is said in line 184 where one can read: "8 different clonal lineages were selected for production of purified antibodies". I suppose it is the 8 antibodies highlighted bold face in Tab. 1 that were recombinantly produced but not all 16 - correct!?

Line 333: consider indicating the other 2 mutations that reduced PC0034 binding in Fig. 9A-9F here in Fig 9I as well - i.e. UL128_K47A and UL131_E23A (maybe in light grey).

Lines 367ff: The authors are correct, that MSL-109 failed in phase II clinical trials. The authors may wish to discuss whether, and if so why, they consider their anti-gH antibodies (which partially overlap with epitopes bound by MSL-109) to be more efficient in clinical trials.

Typos/Formatting

Line 30: interfering "with a post-attachment step" or "with post-attachment steps".

Line 32: "clustered in a similar area or region partially, which are" - consider deleting "partially".

Line 64: consider to change for "Betaherpesvirinae" or "Beta-herpesvirinae"

Line 71: life-threatening "situations" and it seems as if there is a dispensable space character within the brackets "(2, 3)".

Line 96: "gO-null virus" - delete the third "l"

Line 213ff - for my point of view, this sentence requires and additional verb.

Lines 237 and 290: "vise visa" - "vice versa"!?

Line 262: "PC0012 recognizes a much broader region"

Line 263: "the epitope of the well-defined CMV"

Line 311: "bound equally well"

Line 317: "to these 4 pentameric mutants is indeed resulted" - delete "is".

Line 407: "we have further shown" or " we further showed"

Line 572: "followed by PBST" - it seems as if there is a dispensable space character in front of "PBST"

Line 585 should probably read: "/NR or AD169"

Staff Comments:

Preparing Revision Guidelines

Please return the manuscript within 60 days; if you cannot complete the modification within this time period, please contact me. If you do not wish to modify the manuscript and prefer to submit it to another journal, please notify me of your decision immediately so that the manuscript may be formally withdrawn from consideration by Microbiology Spectrum.

Reviewer #1

Point #1: I reviewed an earlier version of this manuscript from Ai and colleagues. Reviewers were impressed by the technical quality of the work, but had concerns about the novelty of results and suggested experiments to address better the mechanism of neutralization. Little to no new experiments were included to address previous reviews. However, changes were made to the writing including more accurate interpretations and presentation of data.

Answer: We are appreciated the reviewer's positive comments on the technical quality of the work and improvement that has been made in the last round of revision. To address better the mechanism of neutralization, we conducted 2 additional experiments to address the concerns of previous reviews. Firstly, we have been attempting to obtain co-crystal structures and perform EM study. Understandably, structural study takes times and sometimes luckiness and we have not yet made breakthrough. Secondly, to evaluate whether group 1 neutralizing antibodies PC0012, PC0014 and PC0035 affect HCMV cell-to-cell transmission after viral attachment, we found that antibodies PC0012, PC0014 and PC0035 along with MSL-109 did not have inhibitory effect on viral spreading in HCMV cell-to-cell spreading assays (Fig 7D) (lines #267-278).

Point #2: I agree that this paper will be of limited use to the field if sequences are not made public. Data can be deposited with GenBank, and authors can request embargo from GenBank until the paper is accepted.

Answer: Following the reviewer suggestions, we have now deposited the mAb sequences to GenBank. Accession numbers for these antibodies were listed in Table 4 (Pages 47).

Point #3: Since specific receptor interactions are not investigated here, I suggest that the detailed receptor information in the Introduction Lines ~ 103-115 be combined with the Discussion.

Answer: Thank you for your suggestions. The detailed receptor information in the Introduction Lines #103-115 has been deleted in the Introduction and has been combined with the Discussion in Lines #414-437.

Point #4: Consider mentioning as data not shown that attempts were made to demonstrate receptor interactions with Nrp2 by biosensor.

Answer: Thank you for your suggestions. The relevant statements in this manuscript have been revised in Line #420-423 as "We have investigated the interaction of pentamer with the relevant receptors by recombinant soluble Nrp2 (M1-P864) with His tag and Nrp2 (M1-P864) with Fc fusion protein. Unfortunately, we failed to detect the binding of Nrp2 to pentamer using the Biacore X100 system and neither was subjected to further analyses".

Reviewer #2

Overall comments: In this manuscript, Ai et al identified a panel of human

monoclonal antibodies against CMV pentameric gH/gL complex from naturally infected donors, and characterized their epitope specificities and abilities for neutralizing CMV infection. Even though no new neutralization mechanism nor binding epitopes were identified, the study does contribute to a growing list of critical resources and tools for the CMV field to better understand the natural CMV immunity, which would potentially inform development of CMV vaccines and therapeutics.

Answer: Thanks to the reviewer's positive comments regarding this work representing a potentially valuable new resource for development of CMV vaccines and therapeutics.

Point #1: I agree with one of the previous reviewers that the sequences of the antibodies need be disclosed, ideally in the manuscript. I also advise the authors to revise the manuscript to more adequately address the reviewers' comments from the previous submission.

Answer: Following this and the reviewer #1's suggestions, we agree and have now deposited mAb sequences to GenBank. Accession numbers for these antibodies were listed in Table 4 (Pages 47). We have made further effort to more adequately address the reviewers' comments from the previous submission.

Point #2: The only additional comment I had is that the authors might want to consider to further characterize the mAbs PC0004, PC0010 and PC0037, and evaluate their ADCC and ADCP activities, or neutralizing potentials in the presence of complement. Any new findings would strengthen the novelty of manuscript. The function of the non-neutralizing antibodies may also play a role in protection. This at least should be discussed.

Answer: Thanks for pointing out and your suggestions. It has been reported that gB non-neutralizing antibodies can also provide protective capacity against HCMV infection (Permar SR et al., 2017; Goodwin ML et al., 2020). There is no report regarding the protective role of non-neutralizing pentamer reactive antibodies. We tested ADCC activity of non-neutralizing pentamer reactive antibodies PC004, PC0010 and PC0037 in ARPE-19 epithelial, but did not identify detectable ADCC activity for these antibodies. The other effector functions such as antibody-dependent cellular phagocytosis (ADCP) or complement dependent cytotoxicity (CDC) of mAbs PC004, PC0010 and PC0037 remain to be proven (Line #397-403).

Reviewer #3

Overall comments: The manuscript entitled "Neutralization epitopes in trimer and pentamer complexes recognized by potent CMV neutralizing human monoclonal antibodies" by Wu et.al describes the characterization of several monoclonal antibodies that bind and/or neutralize diverse strains of human CMV. The studies utilize molecular binding studies as well as virus neutralizing assays to define the specificity of the identified anti-CMV antibodies. Even though previous studies have identified anti-CMV neutralizing mAbs from human B cells, these studies have provided a detailed molecular analysis of mAb binding and neutralization. Importantly, the studies

have defined a neutralizing epitope in the pentamer complex that should be targeted for CMV vaccines.

Answer: Thanks to the reviewer's positive comments regarding this work representing a potentially valuable new resource for development of CMV vaccines and therapeutics.

There are some minor points that should be addressed:

Point #1: Lines 31-36 is unclear and difficult to follow. This should be revised.

Answer: We have now revised and made it more clear for the description in line #32-34 as "These 3 antibodies recognize antigenic epitope(s) clustered in a similar area, which are overlapped with the epitope recognized by the known neutralizing antibody MSL-109."

Point #2: Lines 153-157 are awkward and should be revised.

Answer: Thanks for pointing out. The relevant statements in this manuscript have been revised in Line #142-148 as "Taken together, this study revealed key residues on the antigenic epitopes targeted by these two groups of potent neutralizing antibodies and provided valuable information on neutralizing epitopes on trimer or pentameric complexes for designing and developing trimer and/or pentamer as CMV vaccine. Potent neutralizing mAbs identified in this study can be available as attractive candidates for development of a "cocktail" antibody therapeutic for prevention and treatment of HCMV infection."

Point #3: Line 68 is awkward and should be revised.

Answer: Thanks for pointing out. The relevant statements in this manuscript have been revised using the same numbers "36%" for consistency in Line #21 and #68.

Point #4: Line 237 is awkward and should be revised.

Answer: Thanks for your correction. The line 237 has been corrected on Line #223 and #289 as "vice versa".

Point #5: Line 386 is awkward and should be revised.

Answer: Thanks for pointing out, the description is revised in the paragraph beginning from line #404 as "It is worth noting that neutralizing mAbs PC0012, PC0014 and PC0035 bind to trimer with much higher affinity than to pentamer suggesting that gH/gL epitope on trimer and pentamer is slightly different, but the epitope on trimer may fit better than that on pentamer to these mAbs. It is conceivable that these antibodies were derived from B cells that were initially activated by and matured through the stimulation of the gH/gL epitope on trimer. Conversely, PC0004 bound trimer approximately 20-fold less efficiently than pentamer. It is highly likely that PC0004 antibody was derived from a B cell that was initially activated and matured through by the gH/gL epitope on pentamer. The significance of the difference remains to be elucidated."

Point #6: The generation of pentamer and trimer should include more detail with the transfection conditions to generate these complexes.

Answer: Following your suggestions. The relevant statements in materials and methods have been revised as “Recombinant gH/gL/UL128/UL130/UL131A pentamer was produced in HEK293F by co-transfecting with gH, gL, UL128, UL130 and UL131A plasmids with a mass ratio of 1: 0.8: 0.6: 0.6: 0.6 using the method as described previously (33), while the gH/gL/gO trimer was produced in HEK293F cells by co-transfecting with gH, gL and gO plasmids with a mass ratio of 1: 1: 1.” in Lines #514-518.

Reviewer #4

Overall comments: This manuscript by Ai et al. focuses on the natural humoral immune response directed against its trimer or pentamer complexes. They generated a panel of monoclonal antibodies (mAbs) from healthy individuals and characterized 4 neutralizing mAbs from 2 distinct groups. MAb PC0012, PC0014 and PC0035 in group 1 bind both trimer and pentamer and recognize antigenic epitope(s) clustered in a similar area or region partially overlapped with or close by the epitope recognized by the known neutralizing anti-gH antibody MSL-109. MAb PC0034 from group 2 binds only to pentamer, with binding characteristics similar to the established anti-pentamer antibody 9I6.

Overall, this is a well-conceived study and the data are generally convincing and novel. The results are significant and may contribute to a better understanding of HCMV-entry and how this can be blocked by the humoral immune response against neutralizing epitopes within gH/gL containing complexes.

Answer: Thanks to the reviewer’s positive comments regarding this work representing a potentially valuable new resource for better understanding of HCMV-entry and gH/gL complexes induced humoral immune response.

Minor comments/questions:

Point #1: This study aimed "to identify key protein(s) and antigenic epitopes in HCMV pentamer complex.... for designing better HCMV vaccines.....and provide attractive candidates for the development of an effective cocktail therapeutics" (see lines 52ff). Indeed, the authors identified novel mAbs against established (or partially overlapping) sites in gH and UL128+UL131. However, the most interesting and novel mAb for my point of view is PC0031, which does not compete with any of the established mAbs like MSL-109 or 9I6 or 8I21 but which was unfortunately not further characterized in this study. Since pentamer is considered as an important vaccine immunogen - characterization of the PC0031 epitope might significantly contribute to the overall aim of this study.

Answer: Thank you for your suggestions. MAb PC0031 did not neutralize either CMV Towne strain (Fig. 4A), nor clinical isolate BE13/2012 (data not shown) in fibroblast cell-based neutralization assays. Neutralization assays in epithelial and endothelial cells in this study were performed in collaboration with Prof. Hua Zhu's team at

Rutgers University. Unfortunately, due to the difficulties for transportation of reagents from China to US and limitation of reagents in China during COVID-19 pandemic, PC0031 were not sent for testing at the time. Once we identified neutralizing antibodies among the identified 8 antibodies, we narrowed our focus on characterization for CMV neutralizing antibodies. Therefore, no additional mechanistic studies were carried out for antibody PC0031. We would study PC0031 further when the opportunity becomes available.

Point #2: Similarly, PC0004 has very distinct binding characteristics in Fig.3 but was not characterized in any of the assays Fig.5- Fig.7. The authors should at least provide an explanation for ignoring PC0031 and PC0004.

Answer: Thank you for your suggestions. The reason we did not further characterize PC0004 in Fig.5-Fig.7 was because mAb PC0004 did not neutralize either CMV Towne strain (Fig. 4A), nor clinical isolate BE13/2012 (data not shown). Once we identified neutralizing antibodies among the identified 8 antibodies, we narrowed our focus on characterization for CMV neutralizing antibodies to avoid diluting reader's attention. Regarding PC0031, mAb PC0031 also did not neutralize either CMV Towne strain (Fig. 4A), nor clinical isolate BE13/2012 (data not shown) in fibroblast cell-based neutralization assays. Neutralization assays in epithelial and endothelial cells in this study were performed in collaboration with Prof. Hua Zhu's team at Rutgers University. Unfortunately, due to the difficulties for transportation of reagents from China to US and limitation of reagents in China during COVID-19 pandemic, PC0031 were not sent for testing at the time. Once we identified neutralizing antibodies among the identified 8 antibodies, we narrowed our focus on characterization for CMV neutralizing antibodies. Therefore, no additional mechanistic studies were carried out for antibody PC0031. We would study PC0031 further when the opportunity becomes available. Additional data related to PC0004 and PC0031 were added in Figure 6. Additional data related to PC0004 and PC0031 were included in Figures 6 and 9.

Point #3: Lines 21 and 68: consider to use the same numbers for consistency - i.e. "35%" or "36%".

Answer: Thanks for pointing out. The relevant statements in this manuscript have been revised by using the same number for consistency in Line #21 and #68 as "36%".

Point #4: Line 28: consider to mention where these antibodies were derived from.

Answer: Thank you for your suggestions. The relevant description "we isolated and identified total of 4 neutralizing monoclonal antibodies (mAbs) derived from HCMV-seropositive blood donors." in Lines #26-27.

Point #5: Lines 49f: "...no effective HCMV vaccines or HCMV specific therapeutics have been developed". Consider to formulate this more accurately. This is contrary to what the authors state themselves in lines 119f, 125f or 134-139. In addition, there are

several HCMV specific therapeutics available (GCV, Letemovir, MBV), however their use is limited by toxicity or drug-resistance.

Answer: Thanks for your correction. Description on this has been revised to reflect more precise current status of vaccine and monoclonal therapeutics against CMV infection as “there are still no approved prophylactic vaccines or therapeutic monoclonal antibodies (mAb) for clinical use against HCMV infection although experimental vaccines and therapeutic mAbs have been in clinical trials” on lines #98-101. The description is now consistent to what described in the other places.

Point #6: Line 79: the authors should make clear what is meant with "after the pre-fusion event of the gH/gL complex" - probably "the event" is trimer or pentamer not bound to their receptor(s).

Answer: Thank you for pointing out and suggestion. The description has been revised to make it more clearly “The function of HCMV glycoprotein gB is currently considered to be a major fusogen mediating membrane fusion of virions with infected cells, which is triggered after the trimer or pentamer complex binding to receptors” on lines #77-80.

Point #7: Line 123: "CMVIG exhibit some efficacy against HCMV". The authors may consider to mention recent experience with CMVIG, which showed promising results after SOT (Santhanakrishnan K. et al., 2019; Alsuliman T. et al., 2018; Bonaros N. et al., 2008) or upon biweekly administration efficiently prevented maternal-fetal HCMV transmission after primary infection in the first trimester (Kagan et al., 2019).

Answer: Thanks for pointing out and your suggestions. The description and references “CMVIG shows promising results after SOT or allogeneic hematopoietic cell transplantation (26-28). Upon biweekly administration of CMVIG efficiently prevented maternal-fetal HCMV transmission after primary infection in the first trimester (29).” have been added in lines #106-109.

Point #8: Lines 141-143: Consider to mention all the mAbs that belong to either of the two groups. As far as I understood Fig. 3, PC004, PC0010, PC0037 also belong to group 1, while PC0031 additionally belongs to group 2.

Answer: Thank you for pointing out and your suggestions. The relevant statements have been revised, and PC004, PC0010 and PC0037 in group 1 and PC0031 in group 2 have been included into the relevant description” as below.

Beginning from Line #128, “In this study, we identified and characterized a panel of 8 HCMV pentamer-reactive mAbs that can be divided into 2 groups based on their reactivity to CMV trimer and pentamer. MAbs PC0004, PC0010, PC0012, PC0014, PC0035 and PC0037 in group 1 bind both trimer and pentamer”. Beginning from Line #136, “MAbs PC0031 and PC0034 in group 2 binds only to pentamer. We found one (PC0034) of 2 pentamer-specific mAbs neutralizes HCMV by blocking virus adsorption to the host cells”.

Point #9: Line 153: "studies on the newly identified these two groups of potent" - sounds weird to me.

Answer: Thank you for pointing out. The sentence has been revised as "Taken together, this study revealed key residues on the antigenic epitopes targeted by these two groups of potent neutralizing antibodies and provided valuable information on neutralizing epitopes on trimer or pentameric complexes for designing and developing trimer and/or pentamer as CMV vaccine." in the paragraph beginning from Line #142.

Point #10: Line 170: "(PBMC) of four specimens" - Why does Fig. 2B show only two of them - while Fig. 2A shows all four? Fig 2A, shows high titers of pentamer-reactive antibody responses in Es0099 or Es0055, but no monoclonal antibody against trimer or pentamer could be isolated from these donors?

Answer: The reviewer is correct. It was indeed four specimens with the highest binding antibody titers were selected for sorting single cells. Because antibodies were isolated from the sorted single B cells from only 2 (Es0099 or Es0055) of 4 samples, data for the other 2 samples (ES0055 and ES0099) used for sorting were thus not shown. To make it clearer, we revised legend for Fig. 2B in lines #954-960 as "Shown is the flow cytometry analysis of CMV pentamer-specific memory B cells (inside the red circle) identified by dual color labeled CMV pentamer as a probe by FACS Aria III from 2 (ES0050 and ES0079) of 4 PBMC samples. Data for the other 2 samples (ES0055 and ES0099) were not shown because no CMV pentamer-reactive antibodies were isolated from these donors."

Point #11: Line 199: do the authors have any idea why PC0004 bound trimer 20fold less efficiently than pentamer (Fig 3)?

Answer: Thank you for pointing out. We hypothesize that the gH/gL epitope(s) on trimer and pentamer could be slightly different and the binding properties of a particular antibody might reflect what was the original antigen (gH/gL epitope on trimer or on pentamer) activating the B cells from which the antibody was derived from. In this case, PC0004 antibody could be derived from a B cell activated by and matured through the stimulation of gH/gL epitope on pentamer. Discussion regarding this possibility has been added in Lines #409-412.

Point #12: Given the distinct binding properties of PC0004, wouldn't this mAb be even more important/interesting to include in the assays Fig.4-Fig.7!?

Answer: Reason of the distinct binding properties of PC0004 is discussed in the answer above. Additional data related to P0004 have been added to Figure 4A, Figure 6 and Figure 9. Figure 5 and Figure 7 are for data on comparison of neutralizing antibodies with known-neutralizing antibodies, non-neutralizing antibodies were thus not included.

Point #13: Line 217 and Fig 4B+4C: "...one (PC0034) of them, neutralized HCMV wild strain" - Why do the authors emphasize on this particular mAb? Was PC0031 not tested against all these strains and/or not neutralizing against these strains?

Answer: *Because both PC0031 and PC0034 did not neutralize either CMV Towne strain (Fig. 4A), nor clinical isolate BE13/2012 (data not shown) in fibroblast cell-based neutralization assays, we wanted to know if these antibodies could neutralize HCMV in epithelial and endothelial cells. Neutralization assays in epithelial and endothelial cells in this study were performed in collaboration with Prof. Hua Zhu's team at Rutgers University. Unfortunately, due to the difficulties for transportation of reagents from China to US and limitation of reagents in China during COVID-19 pandemic, PC0031 were not sent for testing at the time. Once we identified neutralizing antibodies among the identified 8 antibodies, we narrowed our focus on characterization for CMV neutralizing antibodies. Therefore, no additional mechanistic studies were carried out for antibody PC0031. We would study PC0031 further when the opportunity becomes available.*

Point #14: Lines 232ff and Fig. 6: "epitope(s) recognized by mAbs PC0012, PC0014 and PC0035 in comparison with the known neutralizing antibody MSL-109" - ... "and the known anti-pentamer antibodies 9I6 and 8I21".

What about PC0031, which is also included in this figure? Isn't it one of the most intriguing findings of this study that PC0031 neither competes with MSL-109 nor the established anti-pentamer mAbs for binding and therefore targets a novel epitope on UL128-131?!?

Answer: *Thank you for point out. Due to the limitation mentioned in the answer above, unfortunately we did not fully study PC0031. Once the opportunity becomes available.*

We would fully characterize PC0031 further in the future study. Additional data for PC0031 was included in Figs.6 and 9E.

Point #15: Line 283ff and Fig. 8: Again, I wonder why PC0031 was not included in these assays. Fig. 6 convincingly demonstrated that PC0034 is a 9I6-like antibody and raised the question to what PC0031 binds. This could have been followed up here in Fig. 8.

Answer: *Thank you for point out. Because our focus on characterization of neutralizing antibodies, once we identified neutralizing antibodies among the identified 8 antibodies, we narrowed our focus on characterization for CMV neutralizing antibodies to avoid diluting reader's attention to include non-neutralizing or unknown antibodies. PC0031 antibody did not cross block with either PC0034 or 9I6. Results related to this is now included in Fig.6.*

Point #16: Line 301: "by blocking the CMV attachment" - consider to change for "binding" to make clear that the mechanism is "post-attachment" as shown in Fig. 7C and written in line 282 of the manuscript.

Answer: *Thanks for your suggestions. The description has been revised as "Thus, these data indicated that mAb PC0034 neutralizes CMV by binding to the pentamer epitope critical for CMV attachment to host cell receptor(s)." in lines #299-301.*

Point #17: Line 312: Consider showing the data because PC0004 has drastically reduced trimer binding characteristics in Fig. 3 and a comparable binding of this antibody could serve as a perfect internal control for complex integrity. In addition, rather than using anti-gH antibodies PC0004 and PC0035 for verification of the integrity of the pentamer complex- I would strongly recommend to utilize PC0031, that belongs to group 2 mAbs and more importantly, which according to Fig6 has significantly different binding characteristics and might therefore be valuable to confirm integrity of the UL-proteins in the pentamer-mutated complexes. It should also be clarified in lines 309f, that PC0004 and PC0035 belong to group 1 mAbs (binding to trimer and pentamer) and are therefore gH/gL-specific.

Answer: Thank you for your suggestions. Binding data of PC0004, PC0035, PC0031 and 9I6 have been added as Fig 9. Text for the results have been added in lines #306-312 as "The Integrity of the mutant pentamer variants was first assessed by ELISA using mAbs PC0004 and PC0035 serving as internal control since these 2 antibodies bind to both trimer as well as pentamer but are not neutralizing. We found that these 11 variants as well as the wild-type pentamer could be bound equally well by both antibodies PC0004 and PC0035 suggesting that there are no apparent quality differences among these 11 pentamer variants (Fig. 9A and 9B). Moreover, Data of 9I6 and PC0031 to bind pentamer and pentamer mutants in ELISA and SPR assays have also been added to Fig.9 and Table 3 (Pages 47).

Point #18: Lines 303: As stated above, Fig. 6 convincingly demonstrated that PC0034 is a 9I6-like antibody. Wouldn't then 9I6 be the perfect control for the experiments with pentamer constructs that harbor mutations in the 9I6 binding site (Fig. 9). Similar to what was done with MSL-109 and its binding to the respective gH mutants in Fig. 7A. This would help to prove or disprove that PC0034 is identical/similar or different to this established mAb 9I6 - and this is one of the main aims of the study!?!?

Answer: Thank you for your suggestions. Data of 9I6 to bind pentamer and pentamer mutants in comparison with PC0034 in ELISA and SPR assays have now been added to Fig.9 and Table 3. Results showed that amino acid residues UL131A_K27A and UL128_T94A strongly impact PC0034 to bind the pentamer with ~81-folds and ~34-folds decrease in affinity, while mutation at these 2 sites had little impact on the binding affinity to 9I6. These results indicate that the PC0034 binding epitope may be slightly different from 9I6.

Point #19: Line 503 "recombinant antibodies were produced in 293T..." - this sentence should be moved after line 507 "Expression and purification of IgG1 recombinant antibody". Was the antibody produced in 293T as stated here or in 293F as written in lines 509 +521?

Answer: Thank you for your suggestions. The Line #503 "recombinant antibodies were produced in 293T..." - this sentence has been moved after Line #556, and word "293F" in line #558 have been revised as "293T". Moreover, these are completely different experimental procedures. First, recombinant antibodies were produced in 293T adherent cells by transient transfection with the purified naturally paired Ig linear

full-length IgG1 and light chain gene expression cassettes, and were then used for screening for binding pentamer antigen in ELISA assay (Fig 2C). Expression constructs were made as plasmids in pCDNA3.1 mammalian expression vector backbone for selected antibodies and used for transfection of 293F suspension cells. Therefore, 293T and 293F cells are used for different purposes and antibodies are produced in small quantities using 293T, while antibodies are produced in large quantities using 293F.

Point #20: Line 512 "16 pentamer-reactive antibodies were cloned...and produced" - this opposes to what is said in line 184 where one can read: "8 different clonal lineages were selected for production of purified antibodies". I suppose it is the 8 antibodies highlighted bold face in Tab. 1 that were recombinantly produced but not all 16 - correct!?

Answer: The reviewer is correct and thanks for pointing out the error. it was indeed that the 8 antibodies highlighted in bold face font in Table 1 were recombinantly produced. The sentence has been revised as "V_HDJ_H and V_LJ_L genes of the full-length IgG1 and light chain genes of the 8 representative pentamer-reactive antibodies as highlighted in bold face in Table 1 were subcloned into pCDNA3.1⁺ (Invitrogen) mammalian expression vector for production antibodies for further characterization." in lines #560-564.

Point #21: Line 333: consider indicating the other 2 mutations that reduced PC0034 binding in Fig. 9A-9F here in Fig 9I as well - i.e. UL128_K47A and UL131_E23A (maybe in light grey).

Answer: Thank you for your suggestions. Two mutations UL128_K47A and UL131_E23A have now been indicated in Fig. 9M and 9N.

Point #22: Lines 367ff: The authors are correct, that MSL-109 failed in phase II clinical trials. The authors may wish to discuss whether, and if so why, they consider their anti-gH antibodies (which partially overlap with epitopes bound by MSL-109) to be more efficient in clinical trials.

Answer: The reason we think the newly isolated antibodies in this study may be better is that among MSL-109-resistant mutants, W168C/R, P171H/S and D446N mutations in gH contributed to the neutralization resistance to MSL-109, mAbs PC0012, PC0014 and PC0035 all bind strongly to pentamer mutants (W168A, P171A and D446A), while mAb MSL-109 did not bind or bound weakly to these mutants as presented in Fig. 7A and discussed in Lines #380-388.

Point #23: Line 30: interfering "with a post-attachment step" or "with post-attachment steps".

Answer: Thanks for your suggestion. The statement has been revised to "... interfering with post attachment steps of CMV entering into cells" in Line #31.

Point #24: Line 32: "clustered in a similar area or region partially, which are" -

consider deleting "partially".

Answer: Thank you for your suggestion. The “region partially” has been deleted in line #33.

Point #25: Line 64: consider to change for "Betaherpesvirinae" or "Beta-herpesvirinae"

Answer: Thanks for your suggestion. Beta herpesvirinae subfamily has been changed to “...beta-herpesvirinae...” in Line #64.

Point #26: Line 71: life-threatening "situations" and it seems as if there is a dispensable space character within the brackets "(2, 3)".

Answer: Thank you for point out. Correction and revision have been made as “... even life-threatening situations (2, 3)” in Lines #71-72.

Point #27: Line 96: "gO-null virus" - delete the third "l".

Answer: Thanks for your correction. The correction has been made in line #96.

Point #28: Line 213ff - for my point of view, this sentence requires and additional verb.

Answer: Thanks for pointing out. The sentence has been revised as “Two pentamer-specific mAbs PC0031 and PC0034 in group 2 did not neutralize either CMV Towne strain (Fig. 4A), nor clinical isolate BE13/2012 (data not shown) in fibroblast cell-based neutralization assays just like the known pentamer-specific neutralizing antibodies 8I21 and 9I6 did” in lines #199-202.

Point #29: Lines 237 and 290: "vise visa" - "vice versa"!?

Answer: Thanks for your correction. The error has been corrected in lines #223 as well as in line #289.

Point #30: Line 262: "PC0012 recognizes a much broader region"

Answer: Thanks for your correction. The relevant statements in this manuscript have been revised in Line #247 as “...PC0012 recognizes”.

Point #31: Line 263: "the epitope of the well-defined CMV"

Answer: Thanks for your correction. The sentence has been revised as “antibody PC0012 recognizes much broader region than do the other 2 antibodies and the well-defined CMV neutralizing antibody MSL-109” in lines #247-249.

Point #32: Line 311: "bound equally well"

Answer: Thanks for your correction. The error has been corrected. “We found that these 11 variants as well as the wild-type pentamer could be bound equally well by 2 pentamer-reactive antibodies” in line #310.

Point #33: Line 317: "to these 4 pentameric mutants is indeed resulted" - delete "is".

***Answer:** Thanks for your correction. The "is" has been deleted in line #316.*

Point #34: Line 407: "we have further shown" or " we further showed"

***Answer:** Thanks for your correction. "We have further shown" haven changed to "we further showed..." revised in Line #454.*

Point #35: "followed by PBST" - it seems as if there is a dispensable space character in front of "PBST"

***Answer:** Thanks for your correction. The extra space has been deleted in line #618-619.*

Point #36: Line 585 should probably read: "/NR or AD169"

***Answer:** Thanks for point out. The correction has been made in line #631 as "NR or AD169 FIX." Moreover, HCMV strain "AD169 FIX" as used throughout the manuscript.*

August 13, 2022

Dr. Hua-Xin Liao
Jinan University
Guangzhou
China

Re: Spectrum01393-22R1 (Neutralization epitopes in trimer and pentamer complexes recognized by potent CMV neutralizing human monoclonal antibodies)

Dear Dr. Hua-Xin Liao:

Thank you for submitting your manuscript to Microbiology Spectrum. Please address the concerns raised by one reviewer and modify your manuscript accordingly. When submitting the revised version of your paper, please provide (1) point-by-point responses to the issues raised by the reviewers as file type "Response to Reviewers," not in your cover letter, and (2) a PDF file that indicates the changes from the original submission (by highlighting or underlining the changes) as file type "Marked Up Manuscript - For Review Only". Please use this link to submit your revised manuscript - we strongly recommend that you submit your paper within the next 60 days or reach out to me. Detailed instructions on submitting your revised paper are below.

Link Not Available

Sincerely,

Haidong Gu

Journals Department
Reviewer comments:

Reviewer #4 (Comments for the Author):

I reviewed an earlier version of this manuscript from Ai and colleagues. The revised manuscript has drastically improved as the authors have answered the majority of my previous concerns and provided reasonable explanations regarding why some experiments were not done (yet), e.g. due to limitations during pandemic. In response to the previous reviewer comments, they also provide interesting novel data, which unfortunately raise novel concerns to me, which should be addressed, and/or conclusions need to be justified before it can be published. I offer my points below for consideration by the authors.

Major comments/questions:

The authors performed an additional experiment from which they concluded that all tested antibodies (except anti-gB) had no

inhibitory effect on HCMV cell-to-cell spread (Fig 7D). This is a rather surprising finding since it is well established that HCMV cell-to-cell transfer is sensitive to antibody-mediated inhibition in epithelial and endothelial cell culture (Gerna et al., 2008, Cui et al., 2013). In addition, a recent publication demonstrated, that neutralizing mAbs against both gB and gH (including MSL-109) could significantly inhibit viral plaque expansion of different HCMV strains (including Towne-GFP) and was equally efficient in fibroblasts as in epithelial cells (Reuter et al., 2022). Wouldn't one, in light of these findings, expect that the neutralizing anti-gH mAbs PC0012, PC0014, PC0035 (Fig 4) with binding characteristics that resemble MSL-109 (Fig. 5, 6 and 7) prevent cell-cell spread like MSL-109 or other anti-gH antibodies!?

Since the authors also used the Towne-GFP virus for their cell-cell spread assay, I would recommend repeating the respective experiments under comparable conditions. This is of particular importance as their novel (MLS-109-like) anti-gH mAbs might have an inhibitory effect on cell-cell spread, if (i) the assay is evaluated later, (ii) performed on epithelial cells and (iii) with higher antibody concentrations. If the novel anti-gH antibodies still do not prevent cell-cell transmission, the authors should provide explanations for potentially discrepant results.

This reviewer totally agrees that non-neutralizing mAbs exert crucial functions *in vivo*, but I would rather like to propose an additional experiment before ultimately concluding that the mAbs PC0004, PC0010 and PC0037 are definitely non-neutralizing. I just realized during this 2nd review that all mAbs were screened for binding to gH of VR1814 (see line 509, Fig 2 and Fig 3) which belongs to gH-genotype gH-1 but the neutralization assay in Fig. 4 was performed with Towne, which belongs to gH-2. Since gH is known to induce strain-specific neutralizing antibody responses (Urban et al. 1992, Ha et al. 2017, Thomas et al. 2021), it might be worth to test PC0004, PC0010 and PC0037 for reactivity (e.g. via FACS or immunofluorescence) and neutralization assays with VR1814-infected cells (like in Fig. 5C). Only if VR1814 is bound but not neutralized, the authors could draw the conclusion on their non-neutralizing activities. My concerns/questions about the ADCC-assay (data not presented) are given below.

Minor comments/questions:

Line 166: "B cells from only 2 of 4 samples" - Fig 2A, shows high titers of pentamer-reactive antibody responses in Es0099 or Es0055, but no monoclonal antibody against trimer or pentamer could be isolated from these donors? The authors may wish to discuss this. Might this be due to the chosen gating strategy (line 543) - can you provide any reference for this gating strategy? Were those donors only recently infected (i.e. IgM positive) and therefore did not contain memory B-cells? Any other explanation?

Line 311-317: Thank you for following my suggestion to utilize PC0004, PC0035 and PC0031 and 9I6 to convincingly confirm the integrity of the UL-proteins in the pentamer-mutated complexes. Congratulations on this beautiful set of data!

Lines 272-281.: "HCMV cell-to-cell spreading assays. HCMV infection was detected through CMV IE1/IE2 expression 4 days after virus inoculation. antibodieshad no inhibitory effect on cell-to-cell spreading of HCMV" - In order to exclude spread via cell-free virus- the authors should consider to (i) use a clinical isolate rather than lab-adapted strain Towne-GFP, (ii) overlay infected cells with methylcellulose/agarose containing up to 50µg/ml but not 10µg/ml of the antibody and most importantly (iii) evaluate not after a single replication cycle but after several days (e.g. see Cui et al., 2017, Murell et al., 2017, Reuter et al., 2022). These facts may explain why especially in case of MSL-109, the authors did not observe an effect on the block of cell-associated spread of CMV in fibroblasts (compare Reuter et al., 2022). In addition, they should consider to repeat the experiment in epithelial cells, where anti-gH mAbs were previously shown to block cell-associated spread (compare Cui et al., 2017).

Line 274+278: "no such small brown particles could be seen" - consider to exchange "particle" as it is likely not a (viral) particle that is stained by the anti-IE1/IE2 antibody but rather ND10/PML-bodies and/or viral replication centers.

Lines 274f: "IE1/IE2-positive plaques of cells newly infected through cell-to cell transmission" - It is not clear to this reviewer, how the authors discriminated between a "newly infected cell" and a neighboring cell which has initiated replication "later/slower", e.g. due to suboptimal derepression of intrinsic immunity?! Why didn't the authors take advantage of the GFP-fluorescence encoded by the utilized recombinant Towne strain and followed the spread of infection over a longer period of time. Plaque size reduction would in this case reveal whether a mAb interferes with cell-associated spread of CMV.

Line 279: "anti-gB neutralizing control antibody" - which antibody was used? Please provide the reference and/or source of this antibody.

Line 307f: "indicated by molecular simulation analysis" - The authors should consider to describe in the methods section how this molecular simulation was performed (PDB, software, parameters, etc.) as this was the basis for the rational design of their 11 pentameric variants.

Line 380: "antibodies in group 1 are neutralizing and cross react with the known gH/gL neutralizing antibody MSL-109" - They do not "cross-react with the MSL-109 antibody" but rather "with an antigen targeted by MSL-109" and/or they "compete with MSL-109 for each other's epitope on gH".

Line 404: "We tested the ADCC activity of the non-neutralizing mAbs PC0004, PC0010 and PC0037....but did not identify detectable ADCC activity" - This reviewer questions this finding. The authors state in lines 556 and 564 that they expressed their recombinant antibodies as IgG1 subtypes, which is known to elicit strong ADCC-responses (<https://www.frontiersin.org/articles/10.3389/fimmu.2020.00740/full>). Did the authors exclude that the Fc-fragment of their IgG-

heavy chains contain any undesired mutations that abrogate ADCC? Did they include some positive control(s) in their assay? Do they have any information about the natural IgG-subtype encoded by the initially isolated B-cell? If so, they should provide this information here.

Line 505. This reviewer would not define AD169 FIX as a HCMV "wild strain" but "laboratory strain with reconstituted pentamer expression". Consider to change or formulate more precisely.

Line 515: "residues 1-715" - The authors may wish to explain the ratio to choose this particular residue. The transmembrane domain seems to start at residue 719 and other groups have chosen amino acid 715 as the terminating residue (and further inserted an SGSG linker between the ectodomain and His-tag) for purification of pentamer (e.g. Chandramouli et al., 2017, PDB 5VOC_A). Why did they chose residue 715 as the terminating residue of the gH-ectodomain and why did they omit the GS-linker?

Line 517: "at 7 different positions " - Again, the authors may wish to explain the ratio for choosing these particular positions and/or reference(s)!

Lines 544ff: The "recombinant pentamer probe was labeled separately with BV421 and PE-Cy7 through biotin-streptavidin conjugation" - The fluorescence was probably coupled to streptavidin - but how was biotin coupled to the purified pentamer from Fig 1A? I suppose via incubation of the purified pentamer with a biotin-coupled anti-His antibody? The authors should explain this critical step and provide the source of this antibody as well as the source for the fluorescence color-labeled antibodies against the cell surface markers.

Lines 602ff: Aliquots of 50µl/well of HCMV Towne... were mixed with 10µl/well antibodies.....after incubation, the mixtures of 100µl/well" - This is 40µl too much for my point of view. Probably it's only 50 or 60µl/well that were added, right!?

Line 608: "cultured ...for 5 days with changing the medium ...on the third day." Why was medium replaced and why was the antibody not replenished? Most importantly, why was Towne evaluated so late in contrast to the clinical isolate BE13/2012 which was already evaluated at 48 hpi (see line 627) as it were the strains VR1814, NR and AD169Fix (line 645), which were stained with an anti-IE1/IE2 antibody? Why was not GFP-fluorescence quantified as in line 723?

Line 614: "known anti-HCMV gB antibody 8F9" - Consider providing the reference and/or source of this particular mAb.

Line 631: "anti-HCMV IE1/IE2 antibody" - Why was this antibody used here and in the following experiment but not in the experiment with Towne, where anti-gB was used?! Is there any particular reason for this?

Lines 706f: DNA isolation was performed at 1h post infection or later?

Line 735: "anti-HCMV IE1/IE2 antibody" - Again the GFP fluorescence encoded by Towne could have been used for quantification of cell-cell spread over the course of several cycles of replication. This might help to explain why the results of this study are in contrast to a recent publication by Reuter et al., 2022, who have observed a significant reduction of cell-associated spread when HFF or ARPE-19 cells were infected with TB40, TR or Towne-GFP in the presence of MSL-109 (and other anti-gH or -gB antibodies). The authors should consider repeating their experiment under comparable conditions and mAb concentrations, include additional negative-control mAbs (like TRN006) and provide explanations for potentially discrepant results. This is of particular importance as it might be that their novel neutralizing anti-gH mAbs might have an inhibitory effect on cell-associated spread - at least in epithelial cells.

Line 1050: "a known anti-fusion positive control (-gB)" - Which antibody was used and do they authors know which of its antigenic domains is targeted!? Overall, this should be formulated more precisely since gB's fusion-activity was not measured with this assay.

Typos/Formatting

Line 200: "0.938 µg/ml, and" - it seems as if there is a dispensable space character in front of "and"

Line 239: "bind gH/gL with in P171" - delete "in"

Line 242: "with signal amino acid substitution" - correct to "single"

Line 254f: "neutralize CMV at post-attachment entry step" - "a post-attachment entry step" or "entry steps"

Line 271: "we evaluate" - exchange for "we evaluated"

Line 298 "endothelia cells" - correct for "endothelial"

Line 389: "in spite that" - "in spite of that"

Line 396: "mAbs PC0012, PC0014 and PC0035 neutralizes" - "neutralize"

Line 458: "neutralize" - "neutralizes"

Line 622: "clinical isolated BE13/2012" - exchange for "isolate"

Lines 680 and 689: "was served as" - delete "was"

Line 686 " anti-his" - consider capitalizing His as in the rest of the manuscript (compare lines 687,690 and 516,753)

Lines 699f + line726: "8x104" - Were the MRC-5 cells seeded with the same density in 24well plates and 96 well plates!??

Line 749: "Hepes" - "HEPES"

Line 1034: "Higher the inhibition percentages are presented as darker" - delete dispensable "the"!?

Fig 4: "(MOI:1)" - delete it from (A) or include the MOIs in (B) and (C) as well.

Fig. 9K +9L+9M: consider to use the same colors for the mutants as in Fig 9A-9E (i.e.: E23A =red, K27A= blue, T94A= purple, avoid green for gH in Fig. 9K)

Fig. 9N: consider to label the red color of UL128_T94 (which is the unique amino acid residue targeted by PC0034) and the pink amino acid UL131A_K27 as it was done for the orange amino acid residues UL128_K47 and UL131A_E23 that might be targeted by both 9I6 and PC0034.

Fig. 9M "Fornt view" - change to "front view", and label the orange residue(s) also here in the front view.

Reference 52 seems to be mixed up. I think [52] should read "Bootz et al, 2017" and SR Permar was not involved in this particular study. In contrast, the study with reference [53] was from the Permar lab. Consider to proof read and correct this.

Staff Comments:

Preparing Revision Guidelines

Please return the manuscript within 60 days; if you cannot complete the modification within this time period, please contact me. If you do not wish to modify the manuscript and prefer to submit it to another journal, please notify me of your decision immediately so that the manuscript may be formally withdrawn from consideration by Microbiology Spectrum.

Corresponding authors may join or renew ASM membership to obtain discounts on publication fees. Need to upgrade your

membership level? Please contact Customer Service at Service@asmusa.org.

Thanks to the reviewer for insightful review and very helpful suggestions. We greatly appreciate the positive comments on the improvement that has been made in the last round of revision of this study. We agree to fully address reviewers' concerns, have performed additional experiments to address your concerns and revised the manuscript accordingly. For your ease in review, the original comments the reviewer are listed before each of our answers in the responses, and major changes or additions are highlighted in the revised manuscript.

Reviewer #4:

Point #1: I reviewed an earlier version of this manuscript from Ai and colleagues. The revised manuscript has drastically improved as the authors have answered the majority of my previous concerns and provided reasonable explanations regarding why some experiments were not done (yet), e.g. due to limitations during pandemic. In response to the previous reviewer comments, they also provide interesting novel data, which unfortunately raise novel concerns to me, which should be addressed, and/or conclusions need to be justified before it can be published. I offer my points below for consideration by the authors.

Answer: We really appreciate the reviewer's positive comments on the improvement that has been made in the last round of revision and insightful review. We agree to make necessary revision throughout the manuscript.

Major comments/questions:

Point #2: The authors performed an additional experiment from which they concluded that all tested antibodies (except anti-gB) had no inhibitory effect on HCMV cell-to-cell spread (Fig 7D). This is a rather surprising finding since it is well established that HCMV cell-to-cell transfer is sensitive to antibody-mediated inhibition in epithelial and endothelial cell culture (Gerna et al., 2008, Cui et al, 2013). In addition, a recent publication demonstrated, that neutralizing mAbs against both gB and gH (including MSL-109) could significantly inhibit viral plaque expansion of different HCMV strains (including Towne-GFP) and was equally efficient in fibroblasts as in epithelial cells (Reuter et al., 2022). Would'nt one, in light of these findings, expect that the neutralizing anti-gH mAbs PC0012, PC0014, PC0035 (Fig 4) with binding characteristics that resemble MSL-109 (Fig. 5, 6 and 7) prevent cell-cell spread like MSL-109 or other anti-gH antibodies!?

Answer: Thanks to the reviewer's comments and suggestion. Following the reviewer's suggestion, we repeated cell-to-cell spread assays and observed the experiments for 5 days and 10 days using the same antibody concentration (50 µg/mL) and assay condition used in the previous studies mentioned by the reviewer. Just as the reviewer expected, we found that mAbs PC0012, PC0014, PC0035 and MSL-109 at 50 µg/mL concentrations inhibited HCMV Towne spreading in MRC-5 cells. Results have now been presented in Fig 7D and described in the lines #273-286.

Point #3: Since the authors also used the Towne-GFP virus for their cell-cell spread assay, I would recommend repeating the respective experiments under comparable

conditions. This is of particular importance as their novel (MLS-109-like) anti-gH mAbs might have an inhibitory effect on cell-cell spread, if (i) the assay is evaluated later, (ii) performed on epithelial cells and (iii) with higher antibody concentrations. If the novel anti-gH antibodies still do not prevent cell-cell transmission, the authors should provide explanations for potentially discrepant results.

Answer: Thank you for your suggestion. We repeated the experiments. See our answers above.

Point #4: This reviewer totally agrees that non-neutralizing mAbs exert crucial functions in vivo, but I would rather like to propose an additional experiment before ultimately concluding that the mAbs PC0004, PC0010 and PC0037 are definitely non-neutralizing. I just realized during this 2nd review that all mAbs were screened for binding to gH of VR1814 (see line 509, Fig 2 and Fig 3) which belongs to gH-genotype gH-1 but the neutralization assay in Fig. 4 was performed with Towne, which belongs to gH-2. Since gH is known to induce strain-specific neutralizing antibody responses (Urban et al. 1992, Ha et al. 2017, Thomas et al. 2021), it might be worth to test PC0004, PC0010 and PC0037 for reactivity (e.g. via FACS or immunofluorescence) and neutralization assays with VR1814-infected cells (like in Fig. 5C). Only if VR1814 is bound but not neutralized, the authors could draw the conclusion on their non-neutralizing activities. My concerns/questions about the ADCC-assay (data not presented) are given below.

Answer: Thank you for your comments and suggestions. As we have stated in our responses for the last round of review that the neutralization assays for VR1814 strain in epithelial and endothelial cells in this study were performed in collaboration with Prof. Hua Zhu's team at Rutgers University. Unfortunately, due to the same reason of difficulties for transportation of reagents from China to US and limitation of reagents in China during COVID-19 pandemic, PC0004, PC0010 and PC0037 were not sent for testing at the time, and we were also not able to test PC0004, PC0010 and PC0037 for reactivity with VR1814-infected cells via FACS or immunofluorescence. Thus we also stated the following possibility: "However, since gH is known to induce strain-specific neutralizing antibody responses (Urban et al. 1992, Ha et al. 2017, Thomas et al. 2021), in spite of that mAbs PC0004, PC0010 and PC0037) did not neutralize HCMV Towne strain, but have not been tested for the ability to neutralize VR1814 strain in epithelial and endothelial cells or to bind the VR1814-infected, It could not be completely rule out the possibility of these 3 antibodies being capable of neutralizing HCMV in epithelial and endothelial cells." in the discussion in lines #411-417.

Also, the relevant statements in discussion have also been revised in line #406 as "We tested the ADCC activity of the HCMV Towne non-neutralizing mAbs PC0004, PC0010 and PC0037 in group I (data not shown), but did not identify detectable ADCC activity for these antibodies. The other effector functions such as antibody-dependent cellular phagocytosis (ADCP) or complement dependent cytotoxicity (CDC) of mAbs PC004, PC0010 and PC0037 remain to be proven.".

Minor comments/questions:

Point #5: Line 166: "B cells from only 2 of 4 samples" - Fig 2A, shows high titers of pentamer-reactive antibody responses in Es0099 or Es0055, but no monoclonal antibody against trimer or pentamer could be isolated from these donors? The authors may wish to discuss this. Might this be due to the chosen gating strategy (line 543) - can you provide any reference for this gating strategy? Were those donors only recently infected (i.e. IgM positive) and therefore did not contain memory B-cells? Any other explanation?

Answer: Thank you for your comments and questions. We sorted single B cells by using dual color fluorescence-labeled pentamer as a probe to avoid sorting non-specific memory B cells as previously described (Ref Morris et al. 2011). When we sorted Es0099 or Es0055 samples, we found that there were unexpected fewer numbers of viable PBMC and more single-color positive B cells than the dual color-positive B cells. These results indicate that cell condition of Es0099 or Es0055 samples might not be in very good shape although Es0099 or Es0055 samples had high serum antibody titers.

Point #6: Line 311-317: Thank you for following my suggestion to utilize PC0004, PC0035 and PC0031 and 9I6 to convincingly confirm the integrity of the UL-proteins in the pentamer-mutated complexes. Congratulations on this beautiful set of data!

Answer: We very much appreciated the reviewer's comments.

Point #7: Lines 272-281.: "HCMV cell-to-cell spreading assays. HCMV infection was detected through CMV IE1/IE2 expression 4 days after virus inoculation. ... antibodies ...had no inhibitory effect on cell-to-cell spreading of HCMV" - In order to exclude spread via cell-free virus- the authors should consider to (i) use a clinical isolate rather than lab-adapted strain Towne-GFP, (ii) overlay infected cells with methylcellulose/agarose containing up to 50µg/ml but not 10µg/ml of the antibody and most importantly (iii) evaluate not after a single replication cycle but after several days (e.g. see Cui et al., 2017, Murell et al., 2017, Reuter et al., 2022). These facts may explain why especially in case of MSL-109, the authors did not observe an effect on the block of cell-associated spread of CMV in fibroblasts (compare Reuter et al., 2022). In addition, they should consider to repeat the experiment in epithelial cells, where anti-gH mAbs were previously shown to block cell-associated spread (compare Cui et al., 2017).

Answer: Thanks to you for insightful comments. Following your suggestions, we have repeated cell-to-cell spreading assays using 50 µg/ml antibody and compatible assay conditions used in the previous studies as you mentioned, just as the reviewer expected, we found that mAbs PC0012, PC0014, PC0035 and MSL-109 at 50 µg/mL concentrations could inhibit HCMV Towne spreading in MRC-5 cells. Results have now been presented in Fig 7D and described in the lines #273-286.

Point #8: Line 274+278: "no such small brown particles could be seen" - consider to

exchange "particle" as it is likely not an (viral) particle that is stained by the anti-IE1/IE2 antibody but rather ND10/PML-bodies and/or viral replication centers.

Answer: Thank you for your suggestion. The "particles" have been revised as "viral replication centers" throughout the manuscript.

Point #9: Lines 274f: "IE1/IE2-positive plaques of cells newly infected through cell-to cell transmission" - It is not clear to this reviewer, how the authors discriminated between a "newly infected cell" and a neighboring cell which has initiated replication "later/slower", e.g. due to suboptimal derepression of intrinsic immunity?! Why didn't the authors take advantage of the GFP-fluorescence encoded by the utilized recombinant Towne strain and followed the spread of infection over a longer period of time. Plaque size reduction would in this case reveal whether a mAb interferes with cell-associated spread of CMV.

Answer: Thank you for your comments and suggestion. Following your suggestions here and comments and question #7 above, we have repeated cell-to-cell spreading assays using 50 µg/ml antibody and followed the spread of infection over a longer period of time (5 and 10 days). Just as the reviewer expected, we found that mAbs PC0012, PC0014, PC0035 and MSL-109 at 50 µg/mL concentrations inhibited HCMV Towne spreading in MRC-5 cells. Results have now been presented in Fig 7D and described in the lines #273-286.

Point #10: Line 279: "anti-gB neutralizing control antibody" - which antibody was used? Please provide the reference and/or source of this antibody.

Answer: Thank you for your question. The anti-gB neutralizing control antibody in Fig 7B to 7D was a gift from Trinomab and was isolated from HCMV serum positive individual. It has been demonstrated as HCMV gB specific neutralizing mAb. High resolution cryo-EM indicates that this mAb targets on antigenic domain 5 (AD-5) on gB glycoprotein. We have now provided the source of this antibody in methods line #730-732.

Point #11: Line 307f: "indicated by molecular simulation analysis" - The authors should consider to describe in the methods section how this molecular simulation was performed (PDB, software, parameters, etc.) as this was the basis for the rational design of their 11 pentameric variants.

Answer: Thank you for your suggestion. The relevant statements in methods section have been revised in line #747-762 as "Structure preparation and MD simulations. The binding mechanism of antibody PC0034 to pentamer was explored using a molecular docking approach.....".

Point #12: Line 380: "antibodies in group 1 are neutralizing and cross react with the known gH/gL neutralizing antibody MSL-109" - They do not "cross-react with the MSL-109 antibody" but rather "with an antigen targeted by MSL-109" and/or they "compete with MSL-109 for each other's epitope on gH".

Answer: Thanks for your correction. The reviewer is correct and relevant statements in

this manuscript have been revised in line #382 as “antibodies in group 1 are neutralizing and compete with the known gH/gL neutralizing antibody MSL-109 on pentamer”.

Point #13: Line 404: "We tested the ADCC activity of the non-neutralizing mAbs PC0004, PC0010 and PC0037....but did not identify detectable ADCC activity" - This reviewer questions this finding. The authors state in lines 556 and 564 that they expressed their recombinant antibodies as IgG1 subtypes, which is known to elicit strong ADCC-responses (<https://www.frontiersin.org/articles/10.3389/fimmu.2020.00740/full>). Did the authors exclude that the Fc-fragment of their IgG-heavy chains contain any undesired mutations that abrogate ADCC? Did they include some positive control(s) in their assay? Do they have any information about the natural IgG-subtype encoded by the initially isolated B-cell? If so, they should provide this information here.

Answer: Thank you for your comments and questions. The Fc-fragment of recombinant mAbs PC0004, PC0010 and PC0037 IgG-heavy chains did not contain any undesired mutations that abrogate ADCC. The positive control antibody was used in ADCC assay contains the same IgG1 Fc-fragment as PC0004, PC0010 and PC0037. Information regarding natural IgG-subtypes encoded by the initially isolated B-cells was included in Table 1.

Point #14: Line 505. This reviewer would not define AD169 FIX as a HCMV "wild strain" but "laboratory strain with reconstituted pentamer expression". Consider to change or formulate more precisely.

Answer: Thanks for your correction. The reviewer is correct and relevant statements in this manuscript have been revised in line #205-207 as “...was demonstrated to neutralize 2 tested HCMV wild strains including VR1814, NR and 1 rescued HCMV strain AD169 FIX (laboratory strain with reconstituted pentamer expression)” and in lines #511 and 642 as “HCMV strains”

Point #15: Line 515: "residues 1-715" - The authors may wish to explain the ratio to choose this particular residue. The transmembrane domain seems to start at residue 719 and other groups have chosen amino acid 715 as the terminating residue (and further inserted an SGSG linker between the ectodomain and His-tag) for purification of pentamer (e.g. Chandramouli et al., 2017, PDB 5VOC_A). Why did they chose residue 715 as the terminating residue of the gH-ectodomain and why did they omit the GS-linker?

Answer: Thank you for your comments and questions. The reviewer is correct and the transmembrane domain of gH was starting at residue 719. The reason we chose residue 715 as the terminating residue of the gH-ectodomain was that we followed a publication by Chandramouli et al that gH gene was terminated at amino acid 715 for gH-ectodomain without the transmembrane domain and cytoplasmic tail (Claudio Ciferri et al., 2017, plos pathogens).

Point #16: Line 517: "at 7 different positions " - Again, the authors may wish to explain the ratio for choosing these particular positions and/or reference(s)!

Answer: Thank you for your question and suggestion. Mutations at 7 different position was made based on the molecular simulations analysis and have now been mentioned in Results.

Point #17: Lines 544ff: The "recombinant pentamer probe was labeled separately with BV421 and PE-Cy7 through biotin-streptavidin conjugation" - The fluorescence was probably coupled to streptavidin - but how was biotin coupled to the purified pentamer from Fig 1A? I suppose via incubation of the purified pentamer with a biotin-coupled anti-His antibody? The authors should explain this critical step and provide the source of this antibody as well as the source for the fluorescence color-labeled antibodies against the cell surface markers.

Answer: Thank you for your comments and suggestion. Indeed, the purified recombinant pentamer was first labeled with biotin using EZ-Link™ Sulfo-NHS-Biotin (thermo #21217). Biotinylated pentamer protein was then coupled with fluorescein dye-labeled streptavidin SA-BV421 or SA-PE-Cy7 and purified. These SA-fluorescein reagents were purchased from Biolegend and have now been described in methods.

Point #18: Lines 602ff: Aliquots of 50µl/well of HCMV Towne... were mixed with 10µl/well antibodies.....after incubation, the mixtures of 100µl/well" - This is 40µl too much for my point of view. Probably it's only 50 or 60µl/well that were added, right!?

Answer: Thanks for your correction. The reviewer is right and the error has now been corrected in line #610 as "the mixtures of 60 µL/well".

Point #19: Line 608: "cultured ...for 5 days with changing the medium ...on the third day." Why was medium replaced and why was the antibody not replenished? Most importantly, why was Towne evaluated so late in contrast to the clinical isolate BE13/2012 which was already evaluated at 48 hpi (see line 627) as it were the strains VR1814, NR and AD169Fix (line 645), which were stained with an anti-IE1/IE2 antibody? Why was not GFP-fluorescence quantified as in line 723?

Answer: Thank you for your comments and questions. In order to maintain the culture of infected cells in healthy condition, we replaced the medium on the third day. In this type of neutralization assays, antibodies were first mixed and incubated with the virus, and the mixtures were then inoculated into cells. Normally, after 2 hours or more of incubation. Inoculum would be removed from the cell cultures without addition of more antibodies. The reason for evaluation of Towne strain was that the detecting antibody used in Cell ELISA method for HCMV Towne neutralization assays was anti-gB antibody 8F9, while gB was late expression viral antigen. In HCMV BE13/2012 neutralization assays, the detecting antibody used in cellular immunostaining method was anti-IE1/2 antibody, while IE1/2 is expressed in the early stage of infection. So, we choose 48h to detect neutralization. As HCMV strains VR1814, NR and AD169 Fix did not have GFP gene, we tested these strains in neutralization assays using similar methods for BE13/2012.

Point #20: Line 614: "known anti-HCMV gB antibody 8F9" - Consider providing the reference and/or source of this particular mAb.

Answer: *Thank you for your suggestion. The human monoclonal antibody 8F9 binds to a linear 10 amino acid epitope of HCMV gB. The relevant reference has been added in the manuscript as reference [66].*

Point #21: Line 631: "anti-HCMV IE1/IE2 antibody" - Why was this antibody used here and in the following experiment but not in the experiment with Towne, where anti-gB was used?! Is there any particular reason for this?

Answer: *Thank you for your suggestion. HCMV strains BE13/2012, VR1814, NR, AD169 FIX did not have GFP gene, thus anti-HCMV IE1/IE2 antibody was used as detecting antibody for cellular immunostaining in HCMV BE13/2012, VR1814, NR, AD169 FIX neutralization assays. HCMV Towne laboratory strain has the GFP gene and multiplies rapidly in MRC-5 cells, so we can assay using GFP fluorescence staining as well as Cell ELISA assays. We used Cell ELISA assay and anti-gB antibody 8F9 to detect HCMV Towne infectivity.*

Point #22: Lines 706f: DNA isolation was performed at 1h post infection or later?

Answer: *Thank you for your question. DNA isolation was performed at 1h post infection. The relevant statements in this manuscript have been revised in line #707 as "Virus DNA isolation was performed at 1h post infection".*

Point #23: Line 735: "anti-HCMV IE1/IE2 antibody" - Again the GFP fluorescence encoded by Towne could have been used for quantification of cell-cell spread over the course of several cycles of replication. This might help to explain why the results of this study are in contrast to a recent publication by Reuter et al., 2022, who have observed a significant reduction of cell-associated spread when HFF or ARPE-19 cells were infected with TB40, TR or Towne-GFP in the presence of MSL-109 (and other anti-gH or -gB antibodies). The authors should consider repeating their experiment under comparable conditions and mAb concentrations, include additional negative-control mAbs (like TRN006) and provide explanations for potentially discrepant results. This is of particular importance as it might be that their novel neutralizing anti-gH mAbs might have an inhibitory effect on cell-associated spread - at least in epithelial cells.

Answer: *Thank you for your comments and suggestions. Following your suggestions, we have repeated cell-to-cell spreading assays using 50 µg/ml antibody and compatible assay conditions used in the previous studies as you mentioned, just as the reviewer expected, we found that mAbs PC0012, PC0014, PC0035 and MSL-109 at 50 µg/mL concentrations inhibited HCMV Towne spreading in MRC-5 cells. Results have now been presented in Fig 7D and described in the lines #273-286.*

Point #24: Line 1050: "a known anti-fusion positive control (α-gB)" - Which antibody was used and do they authors know which of its antigenic domains is

targeted! Overall, this should be formulated more precisely since gB's fusion-activity was not measured with this assay.

Answer: Thank you for your questions and comments suggestion. The anti-gB neutralizing control antibody in Fig 7C to 7D was a gift from Trinomab and was isolated from HCMV serum positive individual. It has been demonstrated as HCMV gB specific neutralizing mAb. High resolution cryo-EM of the complex of this antibody with HCMV gB indicates that this mAb targets on antigenic domain 5 (AD-5) on gB glycoprotein. We have now provided the source of this antibody (Lines #730-732).

Typos/Formatting

Point #25: Line 200: "0.938 µg/ml, and" - it seems as if there is a dispensable space character in front of "and"

Answer: Thanks for your correction. The extra space has been deleted in line #197.

Point #26: Line 239: "bind gH/gL with in P171" - delete "in"

Answer: Thanks for your correction. The "in" has been deleted in line #238.

Point #27: Line 242: "with signal amino acid substitution" - correct to "single"

Answer: Thanks for your correction. The error has been corrected as "single" in line #241.

Point #28: Line 254f: "neutralize CMV at post-attachment entry step" - "a post-attachment entry step" or "entry steps"

Answer: Thanks for your correction. The relevant statement in this manuscript has been revised in line #254 as "entry steps".

Point #29: Line 271: "we evaluate" - exchange for "we evaluated"

Answer: Thanks for your correction. The word has been revised as "evaluated" in line #270.

Point #30: Line 298 "endothelia cells" - correct for "endothelial"

Answer: Thanks for your correction. The error has been corrected as "endothelial" in line #301.

Point #31: Line 389: "in spite that" - "in spite of that"

Answer: Thanks for your correction. The relevant statement has been revised in line #391 as "in spite of that".

Point #32: Line 396: "mAbs PC0012, PC0014 and PC0035 neutralizes" - "neutralize"

Answer: Thanks for your correction. The word has been revised as "neutralize" in line #398.

Point #33: Line 458: "neutralize" - "neutralizes"

Answer: Thanks for your correction. The word has been revised as "neutralizes" in line #466.

Point #34: Line 622: "clinical isolated BE13/2012" - exchange for "isolate"

Answer: Thanks for your correction. The word has been revised as "isolate" in line #626.

Point #35: Lines 680 and 689: "was served as" - delete "was"

Answer: Thanks for your correction. The "was" has been deleted in line #683 and 692.

Point #36: Line 686 " anti-his" - consider capitalizing His as in the rest of the manuscript (compare lines 687,690 and 516,753)

Answer: Thanks for your correction. The relevant statement in this manuscript has been revised in line #522 as "6xHis-tag" and in line #689 as "anti-His-tag".

Point #37: Lines 699f + line726: "8x10⁴" - Were the MRC-5 cells seeded with the same density in 24well plates and 96 well plates!??

Answer: Thanks for your correction. MRC-5 cells were seeded at 4×10⁴ cells per well in 96-well plates and 8×10⁴ cells per well in 24-well plates in this manuscript. The relevant statement in this manuscript has been revised in line #718 and 728.

Point #38: Line 749: "Hepes" - "HEPES"

Answer: Thanks for your correction. The word has been revised as "HEPES" in line #770.

Point #39: Line 1034: "Higher the inhibition percentages are presented as darker" - delete dispensable "the"!?

Answer: Thanks for your correction. The dispensable "the" has been deleted in line #1064.

Point #40: Fig 4: "(MOI:1)" - delete it from (A) or include the MOIs in (B) and (C) as well.

Answer: Thank you for your suggestion. The "MOI:1" has been deleted in Fig 4A.

Point #41: Fig. 9K +9L+9M: consider to use the same colors for the mutants as in Fig 9A-9E (i.e.: E23A =red, K27A= blue, T94A= purple, avoid green for gH in Fig. 9K)

Answer: Thank you for your suggestion. Fig 9A-9E and 9K-9L have been revised (E23A =blue, K27A= pink, T94A= red, K47A= orange, gray for gH in Fig. 9K).

Point #42: Fig. 9N: consider to label the red color of UL128_T94 (which is the unique amino acid residue targeted by PC0034) and the pink amino acid UL131A_K27 as it was done for the orange amino acid residues UL128_K47 and UL131A_E23 that might be targeted by both 9I6 and PC0034.

Answer: Thank you for your suggestion. Fig. 9N has been revised.

Point #43: Fig. 9M "Fornt view" - change to "front view", and label the orange residue(s) also here in the front view.

Answer: Thanks for your correction. The word "Fornt" has been revised as "Front" in Fig. 9M and UL128_K47A has been labeled.

Point #44: Reference 52 seems to be mixed up. I think [52] should read "Bootz et al, 2017" and SR Permar was not involved in this particular study. In contrast, the study with reference [53] was from the Permar lab. Consider to proof read and correct this.

Answer: Thanks for your correction. The reviewer is correct, reference [53] has been revised.

September 23, 2022

Dr. Hua-Xin Liao
Jinan University
Guangzhou
China

Re: Spectrum01393-22R2 (Neutralization epitopes in trimer and pentamer complexes recognized by potent CMV neutralizing human monoclonal antibodies)

Dear Dr. Hua-Xin Liao:

Your manuscript has been accepted, and I am forwarding it to the ASM Journals Department for publication. You will be notified when your proofs are ready to be viewed.

Sincerely,

Haidong Gu
Editor, Microbiology Spectrum
